# PACE: Parameter Change for Unsupervised Environment Design

**Fang Yuan** [1 2 3]  **Junjie Zeng** [1 2]  **Qinglun Li** [1 2]  **Long Qin** [1 2]  **Quanjun Yin** [1 2]  **Siqi Shen** [4]  **Yuxiang Xie** [1]  **Junqiang Yang** [3 2]

## Abstract

Unsupervised Environment Design (UED) offers a promising paradigm for improving reinforcement learning generalization by adaptively shaping training environments, but it requires reliable environment evaluation to remain effective. However, existing UED methods evaluate environments using indirect proxy signals such as regret, value-based errors, or Monte Carlo, which suffer from bias, high variance, or substantial computational overhead. To address these limitations, we propose Parameter Change Environment Design (PACE), a general framework for adaptive level selection in UED. PACE evaluates a level by performing a provisional policy update on it and scoring it with the squared $\ell_2$ norm of the induced parameter change, which directly reflects realized learning progress. This score then guides level selection: levels enter a staleness-aware buffer based on their score, and are replayed via rank-based prioritization, inducing a curriculum that adapts to the agent's evolving capability. By grounding environment evaluation in intrinsic optimization progress, PACE provides a low-variance evaluation signal and avoids the need for additional environment rollouts. Experiments on MiniGrid and Craftax demonstrate that PACE consistently outperforms established UED baselines in zero-shot generalization across diverse out-of-distribution evaluation protocols.

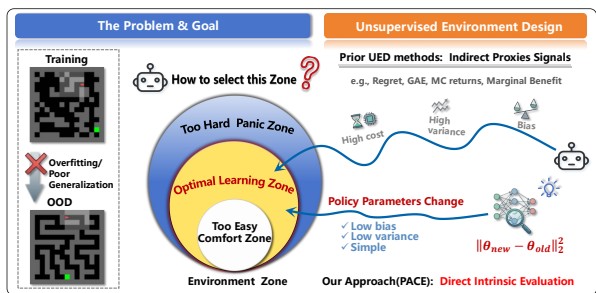

*Figure 1.* UED problem setting and the core idea of PACE. Effective UED requires levels matched to current agent capability, avoiding overly easy or overly hard levels with little learning signal. Existing methods use indirect proxies such as regret, GAE-based errors, Monte Carlo returns, and marginal benefit estimates, which can introduce bias, high variance, or computational overhead. PACE scores a level by the policy parameter change induced by interaction with it, yielding a simple and low-variance signal for realized learning progress.

## 1. Introduction

Reinforcement Learning (RL) (Shakya et al., 2023) achieves remarkable success in complex domains, including game playing (Vinyals et al., 2019; Jain et al., 2024), robotic control (Elguea-Aguinaco et al., 2023; Andrychowicz et al., 2020), and multi-agent systems (Shen et al., 2022; 2023), among many others (Towers et al., 2024). However, RL agents often suffer severe performance degradation when deployed in *out-of-distribution* (OOD) environments, primarily due to overfitting to the training environment distribution. This limitation highlights the central role of training environment design in reinforcement learning, and in particular the need to identify training environments that match the agent current learning capacity and induce effective learning progress.

Unsupervised Environment Design (UED) (Dennis et al., 2020) addresses this challenge by using an underspecified environment to automatically shape the training environment distribution. Training environments are then obtained by assigning values to its free parameters, and we call each resulting instantiated environment a level. In contrast, traditional approaches such as Domain Randomization (DR) (Tobin et al., 2017) sample environment parameters uniformly at random, while minimax adversarial meth-

---

[1]College of Systems Engineering, National University of Defense Technology, Changsha, China [2]State Key Laboratory of Digital Intelligent Modeling and Simulation, Changsha, China [3]Test Center, National University of Defense Technology, Xi'an, China [4]Fujian Key Laboratory of Urban Intelligent Sensing and Computing, Xiamen University, Xiamen, China. Correspondence to: Junjie Zeng <zengjunjie13@nudt.edu.cn>, Qinglun Li <liqinglun@nudt.edu.cn>.

*Proceedings of the $43^{rd}$ International Conference on Machine Learning*, Seoul, South Korea. PMLR 306, 2026. Copyright 2026 by the author(s).

ods (Pinto et al., 2017) generate challenging environments through game-theoretic formulations. However, in practice, both approaches often produce environments with highly inconsistent utility for learning, including instances that are either trivial or effectively unsolvable, which limits their ability to reliably support generalization.

A representative UED method, Protagonist Antagonist Induced Regret Environment Design (PAIRED) (Dennis et al., 2020), evaluates environment value through *regret*, defined as the performance gap between an agent and an optimal policy. Because the optimal policy is generally unavailable, PAIRED relies on a teacher to generate environments and an expert to approximate optimal behavior. To improve efficiency and stability, subsequent works—including Prioritized Level Replay (PLR) (Jiang et al., 2021b), Robust Prioritized Level Replay (PLR$^\perp$) (Jiang et al., 2021a), Adversarially Compounding Complexity by Editing Levels (ACCEL) (Parker-Holder et al., 2022), and Marginal Benefit and Diversity driven Environment Design (MBeDED) (Li et al., 2025)—replace regret with alternative environment evaluation signals. These methods employ proxy metrics such as Generalized Advantage Estimation (GAE) and Monte Carlo (MC) returns, and rely on replay or mutation to construct the training distribution, or measure the return difference between policy snapshots before and after training on a level to quantify marginal benefit.

Despite substantial progress, existing UED methods still lack a reliable learning progress signal. Specifically, regret-based criteria measure theoretical learning potential rather than realized improvement at the current training stage, often selecting levels beyond the current learning capacity. Methods using GAE to approximate regret introduce estimation bias from value function errors, while MC-based methods—whether approximating regret or measuring return differences—reduce this bias at the cost of high variance and substantial computational overhead. As a result, current approaches struggle to provide a signal that simultaneously reflects realized learning progress, maintains low variance, and remains computationally efficient.

We propose **Parameter Change Environment Design** (PACE), a simple and principled framework for environment evaluation in UED. As illustrated in Fig. 1, PACE evaluates the value of a level through the policy parameter change induced by training on it, directly grounding level selection in realized learning progress.

Our contributions are threefold:

- We propose *Parameter Change Environment Design* (PACE), a simple environment evaluation method for UED. PACE measures the level score $S(l)$ using the squared norm of the policy parameter change induced by a provisional update on level $l$. This design provides an intrinsic learning progress signal for level selection, avoids using value-function errors or MC return differences as environment evaluation targets, and requires no additional rollout-based evaluation beyond the trajectory already collected on that level.

- We provide a first-order theoretical analysis that connects parameter change to policy improvement. Under a local gradient-aligned update, we show that the improvement $\Delta J(\theta)$ induced by a level is proportional to the squared $\ell_2$ norm of the corresponding policy parameter update. This proportionality relationship establishes a principled justification for using parameter change as a proxy for environment value.

- We empirically evaluate PACE on MiniGrid and Craftax against established UED baselines. PACE achieves stronger OOD generalization, with an IQM of $0.964$ and an Optimality Gap of $0.172$ on MiniGrid, compared to $0.808$ and $0.299$ for the strongest baseline. On Craftax, PACE attains an IQM of $0.694$ and an Optimality Gap of $0.341$, outperforming all baselines on both metrics.

## 2. Background

In this section, we provide background on UED, including its problem formulation, generalization objective, and representative prior approaches.

### 2.1. Unsupervised Environment Design

Unsupervised Environment Design (UED) studies how to train agents that generalize across a family of environments by actively shaping the distribution of training environments during learning. Following the formulation of (Dennis et al., 2020), UED is formalized as an underspecified partially observable Markov decision process (UPOMDP) $\mathcal{M} = \langle \mathcal{A}, \mathcal{O}, \mathcal{S}, \Phi, \mathcal{T}, \mathcal{I}, \mathcal{R}, \gamma \rangle$, where $\Phi$ denotes a set of free environment parameters and $\mathcal{I}$ is observation function.

Fixing $\phi \in \Phi$ yields a fully specified POMDP $\mathcal{M}_\phi$, which we refer to as a *level* $l$. For a policy $\pi$, its value in level $l$ (equivalently $\mathcal{M}_\phi$) is defined as $V_\phi(\pi) = \mathbb{E}[\sum_{t=0}^{\infty} \gamma^t r_t]$. Because $\Phi$ can be large or continuous, training on all possible levels is infeasible. As a result, the agent can only observe a small subset of levels during learning, which motivates UED methods that select or generate informative levels to support generalization.

### 2.2. The Generalization Objective

Building on the UPOMDP formulation in Section 2.1, each level $l = M_\phi$ is induced by free parameters $\phi \in \Phi$, and the value of a policy $\pi_\theta$ on level $l$ is

$$V_\phi(\pi_\theta) \;=\; \mathbb{E}_{\tau \sim \pi_\theta, \, l} \left[ \sum_{t=0}^{T} \gamma^t r_t \right]. \tag{1}$$

where $\tau$ denotes a trajectory collected under $\pi_\theta$ on $l$, and $\gamma \in (0, 1]$ is the discount factor.

The ultimate goal of UED is to learn a policy that generalizes to unseen levels drawn from a target distribution $P_{\text{test}}$ over $\Phi$. We therefore define the generalization objective as the expected return over $P_{\text{test}}$:

$$J(\theta) \;=\; \mathbb{E}_{\phi \sim P_{\text{test}}}[V_\phi(\pi_\theta)]. \tag{2}$$

In practice, $P_{\text{test}}$ is inaccessible during training. UED instead optimizes a surrogate objective over a training distribution $P_{\text{train}}$, which is dynamically adjusted to approach $J(\theta)$ as the policy evolves. Under this formulation, the central design question becomes: *how to evaluate the training value of each level $l$ with respect to the current policy $\pi_{\theta_{old}}$, so that the resulting curriculum more reliably and effectively drives $P_{train}$ toward levels that most improve generalization.* We review existing answers to this question in Section 2.3, and propose our solution in Section 3.

## 2.3. Existing Approaches to UED

**PAIRED.** PAIRED (Dennis et al., 2020) provides the first principled formulation for UED. Rather than minimizing the return of a single agent, PAIRED introduces an unconstrained antagonist policy that cooperates with the level generator. The generator maximizes *regret*, defined as the performance gap between the antagonist and the protagonist on the same level:

$$\text{REGRET}_\phi(\pi_P, \pi_A) = U_\phi(\pi_A) - U_\phi(\pi_P). \tag{3}$$

where $U_\phi(\pi) = \mathbb{E}_{\tau \sim \pi, \, l(\phi)} \left[ \sum_{t=0}^{T} \gamma^t r_t \right]$ denotes the trajectory return under policy $\pi$. Maximizing regret encourages the generator to produce levels that are challenging yet solvable: the protagonist fails while the antagonist succeeds. As the protagonist improves, the generator constructs increasingly complex levels, inducing an adaptive curriculum aligned with agent capability.

**PLR.** To avoid the cost of training an explicit antagonist, Jiang et al. (2021b) propose PLR. PLR maintains a level buffer $\Lambda$ with size $K$ and stores visited levels with high learning potential. Instead of generating new levels, PLR samples levels from $\Lambda$ according to a trajectory-based score.

The score measures the magnitude of prediction error along a trajectory $\tau$ and uses GAE:

$$S_{\text{PLR}}(\tau) = \frac{1}{T} \sum_{t=0}^{T} \left| \sum_{k=t}^{T} (\gamma\lambda)^{k-t} \delta_k \right|. \tag{4}$$

where $\delta_k = r_k + \gamma V(s_{k+1}) - V(s_k)$ is the temporal-difference error.

**Robust PLR.** To strengthen the link between replay and regret, Jiang et al. (2021a) reinterpret PLR under minimax regret and propose Robust PLR (PLR$^\perp$). Robust PLR suppresses policy updates on newly sampled levels and trains only on replayed levels from $\Lambda$. For regret-based prioritization, it uses *Positive Value Loss* (PVL), which retains only positive GAE values:

$$S_{\text{PLR}}^{\perp}(\tau) = \frac{1}{T} \sum_{t=0}^{T} \max\left( \sum_{k=t}^{T} (\gamma\lambda)^{k-t} \delta_k, \, 0 \right). \tag{5}$$

It also considers Maximum Monte Carlo (MaxMC), which scores level $l$ as $S_{\text{MaxMC}}(\tau) = \frac{1}{T} \sum_{t=0}^{T} \left( R_{\max}(l) - V(s_t) \right)$. by replacing bootstrapped value targets with the best observed return, reducing bias from current value estimates.

**ACCEL.** While PLR replays informative past levels, ACCEL (Parker-Holder et al., 2022) emphasizes explicit difficulty control. ACCEL samples levels from a random generator and applies targeted edits to maintain a desired agent success rate. This mechanism keeps training levels near the agent performance frontier while avoiding levels that are excessively hard or trivial.

**MBeDED.** Li et al. (2025) propose Marginal Benefit as an agent-centric criterion that directly measures realized improvement. Given a baseline policy $\pi_B$ and an updated policy $\pi_A$, the marginal benefit of level $l$ is $\mu_\phi(\pi_A, \pi_B) = V_\phi(\pi_A) - V_\phi(\pi_B)$. By focusing on performance improvement rather than static difficulty, this metric better aligns level selection with learning progress. However, it still requires additional rollouts and Monte Carlo estimation, introducing nontrivial computational overhead and variance.

While existing methods advance Unsupervised Environment Design from different perspectives, their environment evaluation metrics still suffer from fundamental limitations. PAIRED maximizes regret, which reflects the theoretical learning potential of a level rather than the realized capability of the current agent at its training stage. As a result, it tends to select levels that exceed the current learning capacity and yield limited practical improvement. Replay-based methods such as PLR and ACCEL show strong empirical performance under random environment generation, but their evaluation metrics remain imperfect. Specifically, methods using GAE to approximate regret introduce estimation bias from value-function errors. MC-based metrics avoid this bias but introduce high variance, while MBeDED further amplifies both variance and overhead by comparing MC returns from different policy snapshots.

Overall, existing UED methods either rely on proxy signals weakly correlated with realized learning progress, or incur substantial estimation variance and computational overhead. This motivates the need for a simple, low-variance, and intrinsic learning progress signal for level selection.

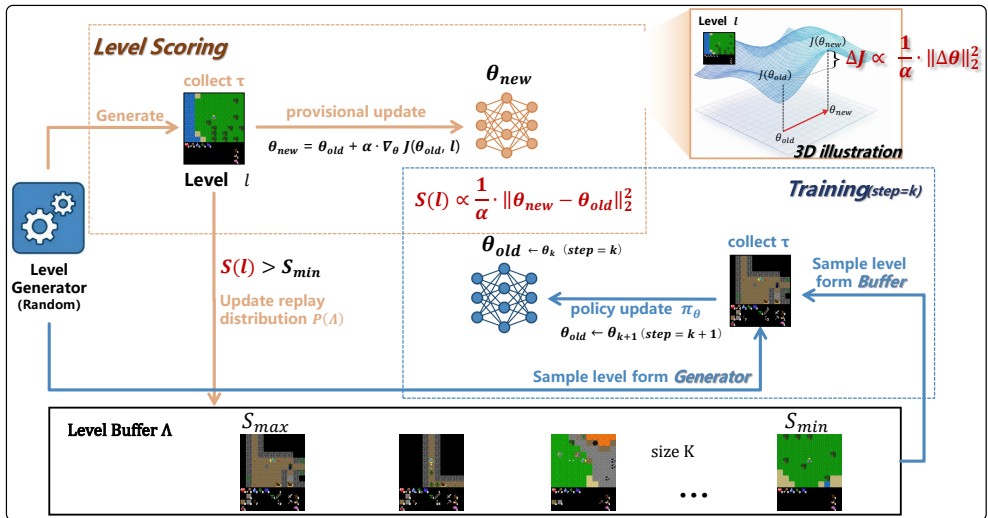

*Figure 2.* Overview of PACE. At training step $k$, the current policy is $\pi_{\theta_{\text{old}}}$ with $\theta_{\text{old}} = \theta_k$. PACE separates level scoring (orange) from policy training (blue). In the scoring stage, a random generator proposes a level $l$, the agent collects a trajectory $\tau$ under $\pi_{\theta_{\text{old}}}$, and a provisional update produces temporary parameters $\theta_{\text{new}}$. The 3D illustration visualizes the objective landscape induced by level $l$ and shows the mapping from induced policy parameter change to an estimate of objective improvement. Motivated by the first-order analysis in Section 3.1, PACE uses the induced parameter change to define the level score $S(l)$ in Eq. 11, without writing the provisional update back to the policy. The level is added to the level buffer $\Lambda$ if its score is higher than the lowest stored score. In the training stage, PACE samples levels from $\Lambda$ using the replay distribution in Eq. 12 and updates the policy from $\pi_{\theta_k}$ to $\pi_{\theta_{k+1}}$.

## 3. Approach

In this section, we introduce **Parameter Change Environment Design (PACE)**, a UED method that uses policy parameter updates as a simple and intrinsic proxy for agent learning progress. In Section 3.1, we formalize environment value from the perspective of performance improvement and provide the corresponding theoretical derivation. Section 3.2 then describes how to select and reuse high-value environments in practice.

### 3.1. Parameter Change as a Signal of Learning Progress

Let $\pi_{\theta_{\text{old}}}$ denote the current policy before interacting with a level $l$. After collecting trajectories $\tau$ in $l$ and applying a single local policy update, the policy parameters change from $\theta_{\text{old}}$ to $\theta_{\text{new}}$, yielding a new policy $\pi_{\theta_{\text{new}}}$. We characterize the learning progress induced by level $l$ through the improvement in the policy optimization objective, and use this quantity to define the environment value in UED:

$$\Delta J(l) = J(\theta_{\text{new}}, l) - J(\theta_{\text{old}}, l). \quad (6)$$

In reinforcement learning, evaluating $J(\theta, l)$ relies on stochastic return estimation. Directly estimating $\Delta J(l)$ therefore requires additional environment interactions and introduces substantial estimation noise. We instead seek a deterministic and intrinsic approximation of $\Delta J(l)$ that depends only on the policy update induced by the level.

We derive this approximation under an idealized local anal-

ysis with a single gradient-based update. For a fixed level $l$, consider the objective $J(\theta, l)$ and apply a first-order Taylor expansion around $\theta_{\text{old}}$:

$$J(\theta_{\text{new}}, l) \approx J(\theta_{\text{old}}, l) + \nabla_\theta J(\theta_{\text{old}}, l)^\top (\theta_{\text{new}} - \theta_{\text{old}}). \quad (7)$$

Assume a single-step update that follows the local gradient of the optimization objective,

$$\theta_{\text{new}} = \theta_{\text{old}} + \alpha \nabla_\theta J(\theta_{\text{old}}, l). \quad (8)$$

where $\alpha$ denotes the step size. Rearranging yields

$$\nabla_\theta J(\theta_{\text{old}}, l) = \frac{1}{\alpha}(\theta_{\text{new}} - \theta_{\text{old}}). \quad (9)$$

Substituting into the Taylor expansion gives a first-order approximation of the objective improvement:

$$\begin{aligned}
\Delta J(l) &= J(\theta_{\text{new}}, l) - J(\theta_{\text{old}}, l) \\
&\approx \nabla_\theta J(\theta_{\text{old}}, l)^\top (\theta_{\text{new}} - \theta_{\text{old}}) \\
&= \frac{1}{\alpha}(\theta_{\text{new}} - \theta_{\text{old}})^\top (\theta_{\text{new}} - \theta_{\text{old}}) \\
&= \frac{1}{\alpha}\|\theta_{\text{new}} - \theta_{\text{old}}\|_2^2.
\end{aligned} \quad (10)$$

This analysis highlights a key geometric relationship: under a local gradient-aligned update, the improvement in the optimization objective induced by a level is proportional to the squared $\ell_2$ norm of the resulting policy parameter change, up to a constant factor. Appendix B.1 further analyzes the

omitted second-order term and the corresponding conditions for local validity.

We emphasize that this derivation relies on idealized assumptions and serves as theoretical motivation rather than a precise characterization of the optimization dynamics used in practice. In Appendix B.2, we further analyze the PACE score under stochastic gradient estimates and discuss how finite-sample gradient estimation issues (Ilyas et al., 2018) affect the validity of these assumptions in practice. Despite these simplifications, the first-order relationship motivates the use of policy parameter change as a simple and intrinsic proxy for environment value, which we operationalize in the following section.

### 3.2. PACE Algorithm

Based on the above analysis, we propose Parameter Change Environment Design (PACE), a UED framework that uses policy parameter change as an intrinsic signal of realized learning progress, which is then used to construct environment value for level selection. An overview of the framework is illustrated in Figure 2.

First, PACE evaluates levels by collecting a trajectory $\tau$ on a newly generated level $l$ and performing a provisional policy update for scoring purposes. In this framework, we adopt a simple random distribution as the level generator, which provides a diverse set of environments while remaining conceptually and computationally lightweight.

We then compute the level score according to Eq. 11, which measures the magnitude of the induced policy parameter change. Based on this score, the level is inserted into the level buffer $\Lambda$ by replacing the lowest-scoring level when the buffer is full. As a result, this procedure ensures that $\Lambda$ consistently retains levels that induce substantial policy parameter updates. The complete procedure is summarized in Algorithm 1.

$$S(l) = \frac{1}{\alpha}\|\theta_{\text{new}} - \theta_{\text{old}}\|_2^2. \tag{11}$$

In addition, as illustrated in Figure 2, PACE samples levels from the level buffer $\Lambda$ during training using a score-based prioritized distribution. Following PLR (Jiang et al., 2021b), we adopt rank-based prioritization, where levels are ranked by their scores and sampled according to

$$P(l_i) = \frac{\left(\frac{1}{\text{rank}(S(l_i))}\right)^{1/\beta}}{\sum_{l_j \in \Lambda}\left(\frac{1}{\text{rank}(S(l_j))}\right)^{1/\beta}}, \tag{12}$$

where $\beta$ is a temperature parameter that controls the extent to which the inverse rank $\frac{1}{\text{rank}(S(l_i))}$ influences the resulting sampling distribution.

At the same time, as training progresses, the replay distribution may gradually concentrate on a subset of frequently sampled levels. Levels that are not revisited for extended periods risk being neglected, even if they were previously informative. Such imbalance can lead to catastrophic forgetting, where the agent loses proficiency on under-sampled levels. To mitigate this effect, we incorporate a staleness-aware mechanism from PLR (Jiang et al., 2021b) that encourages periodic re-evaluation of stored levels. By promoting the replay of levels that have not been sampled recently, this mechanism helps maintain coverage of the level buffer and stabilizes training over time.

**Strength of PACE.** By grounding environment evaluation in policy updates, PACE offers three advantages over existing UED approaches.

**First**, PACE aligns environment selection with realized learning progress. The first-order analysis in Section 3.1 shows that $S(l)$ directly reflects how much a level $l$ contributes to the policy optimization objective $\Delta J(l)$, rather than to theoretical solvability gaps or prediction errors. A formal comparison with loss-based scoring signals in the PLR family is provided in Appendix B.3.

**Second**, PACE provides a low-variance evaluation signal, since $S(l)$ depends solely on the $\ell_2$ norm of the induced parameter update and avoids the sampling variance incurred by MC return estimation. This property is further confirmed by the coefficient of variation (CV) (Abdi, 2010) analysis on MiniGrid and Craftax in Section 4.3.

**Third**, PACE is computationally efficient. For each level score, scoring requires only a provisional policy update whose parameters are discarded after computing $S(l)$, without any additional policy re-evaluation or rollback rollout. A detailed complexity analysis is provided in Appendix B.4.

## 4. Experimental results

In this section, we present experimental results on the *MiniGrid* and *Craftax* benchmarks to demonstrate zero-shot generalization when a trained agent transfers to OOD environments. We compare **PACE** against several representative UED baselines, including DR (Tobin et al., 2017), PLR (Jiang et al., 2021b), PLR$^\perp$ (Jiang et al., 2021a), ACCEL (Parker-Holder et al., 2022). PAIRED and MBeDED rely on environment generation through an RL teacher, which introduces additional policy instances and environment interactions and leads to a substantially different computational model. Therefore, we leave a systematic comparison with these methods to future work. Nevertheless, we present a detailed analysis of computational complexity and evaluation stability in Appendix B.4.

In all cases, we train a student agent using Proximal Policy

**Algorithm 1** PACE

---

**Input:** Level buffer size $K$, initial fill ratio $\rho$, replay
probability $p$, replay distribution $P$, level generator
**Initialize:** Policy $\pi_{\theta_{old}}$, level buffer $\Lambda$
Sample $K \cdot \rho$ initial levels to populate $\Lambda$
**while** training not converged **do**
    Sample replay decision $\epsilon \sim \mathcal{U}[0, 1]$
    **if** $\epsilon \geq p$ **then**
        Sample a new level $l$ from the level generator
    **else**
        Sample level $l$ from $\Lambda$ according to $P$
        Update policy to $\pi_\theta$ on level $l$
    **end if**
    Collect trajectory $\tau$ on level $l$
    Provisional update to obtain $\theta_{new}$ for scoring
    Compute level score $S(l)$ (Eq. 11)
    **if** $S(l)$ exceeds the lowest score in $\Lambda$ **then**
        Update $\Lambda$ with level $l$
    **end if**
    Set $\theta_{old} \leftarrow \theta$
**end while**

---

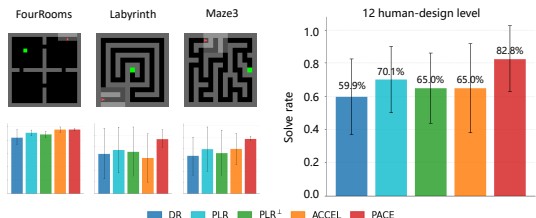

*Figure 3.* Zero-shot transfer on human-designed MiniGrid levels. The first three panels show solve rates on representative OOD levels of increasing difficulty; the rightmost panel reports the average across all 12 levels. Bars denote mean and standard deviation over 10 training seeds. PACE achieves higher solve rates with smaller standard deviation than baselines.

Optimization (PPO) (Schulman et al., 2017) with the same policy architecture and optimizer. A complete list of hyperparameters for each experiment is provided in Table 9 in Appendix C. We show all performance metrics as a function of the number of gradient updates applied to the student policy. The corresponding total number of environment interactions is reported in Table 2 in Appendix C. We implement all experiments using JaxUED (Coward et al., 2024), a publicly available JAX-based framework for UED. MiniGrid experiments run on a single RTX 5090 GPU (32G), while Craftax experiments run on a single A100 GPU(80G).

We begin with a partially observable navigation environment and evaluate transfer performance on human-designed levels in MiniGrid (Chevalier-Boisvert et al., 2018). Then, We compare performance in Craftax (Matthews et al., 2024), a JAX-based benchmark for open-ended reinforcement learning with substantially more complex dynamics, including multi-floor worlds, diverse enemies, and long-horizon objectives. Craftax builds upon the Crafter environment (Hafner, 2021) and extends it with mechanics inspired by Roguelike games and NetHack (Küttler et al., 2020).

### 4.1. MiniGrid Environments

The MiniGrid environment is widely used in UED research. It supports rapid construction of diverse levels by adjusting parameters such as the number of obstacles and maze size, which enables flexible control over task difficulty. Although the overall task structure is relatively simple, training robust agents typically requires large-scale training. Therefore, in this experiment, we train the student agent for 30k policy

updates, which corresponds to 245,760,000 environment interaction steps. During training, all training levels use mazes with a size of $15 \times 15$ tiles. For all DR-based methods, we generate training levels by performing 100 random obstacle placement operations. In contrast, for ACCEL, we begin with empty rooms and randomly edit block locations by adding or removing blocks, as well as modifying the goal location. To improve computational efficiency, we run 32 parallel environments. For each level $l$, the score $S(l)$ is computed from a 256-step rollout collected on that level.

We follow definitions from prior work when selecting environment evaluation metrics. Specifically, PLR uses L1 value loss, following (Jiang et al., 2021b). PLR$^\perp$ and ACCEL adopt MaxMC as metric, following (Jiang et al., 2021a; Parker-Holder et al., 2022). We evaluate zero-shot transfer performance on 12 human-designed levels, with a maximum episode length of 250 steps. The detailed levels can be found in Figure 11 in Appendix C.1. All methods use 10 random seeds for training, and we evaluate the policy obtained at the end of PPO training. All experimental settings match those in prior work.

In Figure 3, we select three representative OOD test levels with progressively increasing difficulty and report the mean and standard deviation of solve rates for each method on these levels. We observe that on the relatively easy FourRooms level, all methods achieve solve rates above 84%, indicating that this class of simple OOD environments poses little challenge for most approaches. However, as level complexity increases, performance differences among methods become increasingly pronounced. On the more challenging Labyrinth and StandardMaze3 levels, PACE consistently achieves substantially higher and more stable solve rates than the baselines. In particular, PACE maintains solve rates above 90% (0.943 on Labyrinth and 0.983 on Maze3) with markedly smaller standard deviation (0.150 and 0.042, respectively), demonstrating stronger robustness in complex OOD environments.

In addition, the right panel of Figure 3 summarizes the

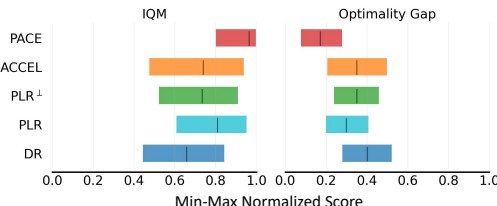

*Figure 4.* Aggregate zero-shot performance on 12 MiniGrid levels, measured by IQM (higher is better) and Optimality Gap (lower is better). PACE attains the best values on both metrics with tighter intervals than all baselines.

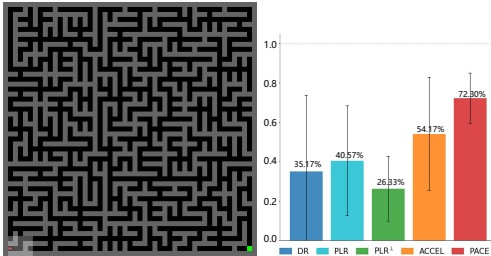

*Figure 5.* Zero-shot generalization to the large-scale PerfectMaze environment. PACE achieves a 72.3% solve rate, substantially outperforming all baselines. Bars show mean and standard deviation over 10 training seeds.

average solve rate and standard deviation across 12 human-designed test levels. Overall, PACE improves the average solve rate by more than 12% over the baselines and attains the smallest standard deviation (0.198, compared to 0.202–0.27 for the baselines), indicating more stable and consistent generalization across OOD levels with varying degrees of complexity. Complete per-level results appear in Figure 12 and Table 3 in Appendix C.1.

To evaluate generalization across OOD levels with varying complexity from a distributional perspective, we adopt the Interquartile Mean (IQM) and Optimality Gap metrics from the `rliable` (Agarwal et al., 2021) library to summarize the distribution of solved rates across methods. Figure 4 reports the corresponding results. PACE achieves an IQM of 96.4% in terms of solved rates, which is substantially higher than all baseline methods. The strongest baseline, PLR, attains an IQM of 80.8%. Inspection of the IQM confidence intervals shows that PACE consistently maintains stronger performance in the central region of the solved-rate distribution across different OOD levels. Under the Optimality Gap metric, PACE also achieves the smallest gap, with a value of 17.2%, which is notably lower than the baselines. Complete numerical results and statistical significance tests appear in Table 4 in Appendix C.1.

Finally, we evaluate zero-shot transfer performance on the largest version of PerfectMaze, a procedurally generated maze environment, as shown in Figure 5. This setting contains levels with 51×51 tiles and a maximum episode length

of 3k steps, which is substantially larger than the environments used during training. We evaluate agents for 100 episodes per training seed, using the same policy checkpoints as in Figure 3. As shown in Figure 5, PACE significantly outperforms all baseline, achieving a solve rate of 72.3%. These results further demonstrate that PACE generalizes effectively to OOD environments with substantially increased scale and complexity.

## 4.2. Craftax Environments

Additionally, we further evaluate PACE in Craftax, which induces a continually expanding and non-stationary task distribution as new regions, mechanics, and objectives emerge over time (Matthews et al., 2024). This setting tests whether UED methods maintain robust performance under long-horizon exploration and continual changes in task structure. Following the Craftax-1B Challenge (Matthews et al., 2024), we evaluate agents on Craftax-Symbolic under a budget of 1 billion environment interactions, where 1B timesteps correspond to approximately six years of human gameplay. We implement this budget by training the student agent for 255 policy updates. Similar to MiniGrid, we use 1,024 parallel environments for efficient data collection. Since Craftax involves substantially longer horizons and richer non-stationary dynamics, we use a longer effective scoring horizon of 4,096 transitions per level, implemented as 64 consecutive 64-step rollout segments on the same level.

We compare PACE with DR and established UED baselines, including PLR, PLR⊥, and ACCEL. Specifically, PLR uses L1 value loss, while PLR⊥ and ACCEL use the PVL metric. For ACCEL, we follow the Craftax-1B setup (Matthews et al., 2024) and adopt the noise mutation operator, where each level $l$ is parameterized by the angle vectors of the Perlin noise generator and mutation adds uniform noise to each element. This variant is reported as the strongest ACCEL variant in prior work.

We report training reward as a function of environment interactions as Figure 6. In the early stage of training, all methods achieve similar reward, while differences become more pronounced as training progresses. As training proceeds, PACE continues to improve steadily in the later phase of training and finishes with the highest reward among all compared methods, indicating more sustained progress under the long-horizon and non-stationary dynamics of Craftax.

After training for 1 billion timesteps, we evaluate the final saved checkpoints on a fixed set of 20 evaluation levels following the Craftax-1B protocol (Figure 7). PACE achieves the highest evaluation reward, with a mean reward of $31.198 \pm 2.439$ over 10 training seeds, outperforming all baselines under the same interaction budget. We provide per-level results across the 20 evaluation maps in Figure 13 and Table 5 in Appendix C.2, showing that the performance

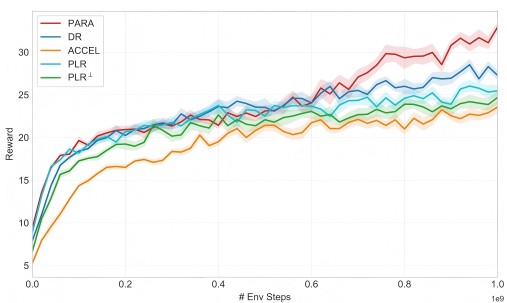

*Figure 6.* Training reward on Craftax-1B as a function of environment interactions. Curves show the mean over 10 training seeds, and shaded regions denote 1 standard error.

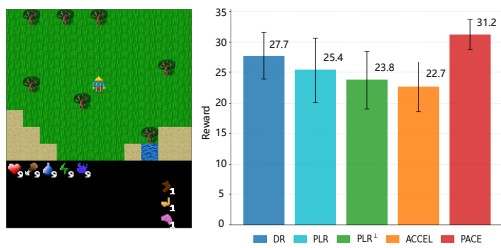

*Figure 7.* Evaluation reward on the Craftax-1B test set after training. Bars show the mean and standard deviation over 10 seeds.

advantage is not driven by a small subset of favorable levels.

We also apply the same IQM and Optimality Gap metrics to summarize reward distribution across evaluation levels and seeds. Figure 8 reports the corresponding results with confidence intervals. PACE achieves the highest IQM (0.694; CI: [0.532, 0.850]) among all methods, while the strongest baseline under IQM, PLR$^\perp$, attains 0.618 (CI: [0.361, 0.832]). Under the Optimality Gap metric, PACE also yields the smallest gap (0.341; CI: [0.209, 0.473]), improving over all baselines. Complete numerical results are provided in Table 6 in Appendix C.2.

These results also suggest that realized learning progress becomes increasingly important as environment complexity grows. In Craftax, earlier UED methods bring limited gains. Unlike the gridworld mazes on which they have been tested, Craftax consistently generates high-quality, solvable levels with rich open-ended structure (Matthews et al., 2024). As a result, the agent is not penalized by training on DR levels, which may explain why regret-based curation methods such as ACCEL yield limited gains over DR in this setting. As training progresses, however, the increasing complexity and non-stationarity of Craftax make it more important to identify the frontier where the agent can still make meaningful progress. PACE directly aligns environment selection with realized learning progress, which helps maintain a training distribution matched to evolving agent capability and leads to stronger generalization across evaluation levels.

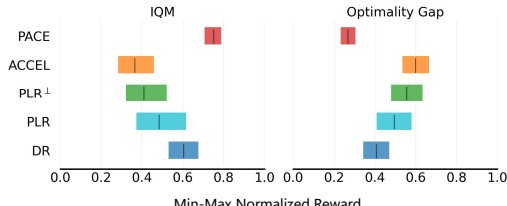

*Figure 8.* Aggregate OOD performance on Craftax, measured by IQM and Optimality Gap over Craftax evaluation levels. Higher IQM and lower Optimality Gap indicate better performance.

### 4.3. Reliability of the Scoring Signal

**Score Variability.** We examine whether the scoring signal yields stable estimates by directly targeting the level score $S(l)$, rather than downstream solved rates or episodic rewards. Let $\pi_{\theta_i}$ denote the policy at the end of training for checkpoint $i$, and $l_j$ a fixed level. For each pair $(\pi_{\theta_i}, l_j)$, we compute the score $R$ times with independently sampled trajectories $\tau$ under the same $\pi_{\theta_i}$,

$$\hat{S}^{(r)}(l_j; \pi_{\theta_i}), \quad r = 1, \ldots, R. \tag{13}$$

where PACE applies the parameter-change score in Eq. (11), and PLR, PLR$^\perp$, ACCEL follow the scoring rules adopted in our main experiments.

Since these methods adopt different scoring rules, the resulting scores differ in magnitude, and a direct comparison of standard deviations is dominated by scale differences. We therefore adopt the coefficient of variation (CV) (Abdi, 2010), which normalizes the standard deviation by the mean and yields a scale-invariant measure of relative dispersion, with the detailed computation provided in Appendix C.3.

Based on this metric, Table 1 reports the resulting Mean CV. PACE attains the lowest score-estimation variability on both benchmarks, empirically validating $S(l)$ as a low-variance signal for environment value. Detailed per-level results are further provided in Tables 7 and 8 in Appendix C.3.

*Table 1.* Stability of the score signal under fixed policy-level pairs, measured by Mean CV. MiniGrid levels span different difficulty ranges drawn from the 12 held-out levels, while Craftax levels are randomly generated and fixed across all methods. Lower is better.

| Environment | Method | #Checkpoints | #Levels | #Repeats | Mean CV ↓ |
|---|---|---|---|---|---|
| MiniGrid | PLR | 5 | 5 | 30 | 0.0079 |
| | PLR$^\perp$ | 5 | 5 | 30 | 0.0055 |
| | ACCEL | 5 | 5 | 30 | 0.0135 |
| | PACE | 5 | 5 | 30 | **0.0010** |
| Craftax | PLR | 10 | 5 | 30 | 0.6105 |
| | PLR$^\perp$ | 10 | 5 | 30 | 0.6926 |
| | ACCEL | 10 | 5 | 30 | 0.6257 |
| | PACE | 10 | 5 | 30 | **0.5412** |

**Score Correlation.** We further examine whether the PACE score is empirically associated with reward improvement during training. For checkpoint $i$ and level $l$, we mea-

sure the reward improvement between consecutive checkpoints as

$$\Delta R_i(l) = R(\pi_{\theta_{i+1}}, l) - R(\pi_{\theta_i}, l). \qquad (14)$$

Figure 9 plots PACE score against reward improvement on Craftax. The two quantities exhibit a positive correlation, with Spearman $\rho = 0.641$ and $p = 1.4 \times 10^{-39}$. This result suggests that the score aligns with actual learning progress.

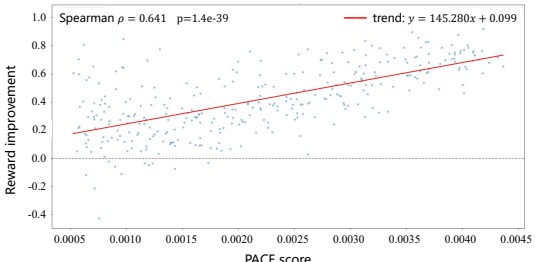

*Figure 9.* Scatter plot of score versus reward improvement on Craftax across 10 seeds. The red line shows a linear regression fit.

## 5. Related Work

In this section we provide a brief overview of related work. A more comprehensive discussion appears in Appendix A.

A central challenge in deep RL is poor generalization to out-of-distribution environments (Amir et al., 2024; Huang et al., 2025; Zhang et al., 2021). Domain Randomization (DR) (Tobin et al., 2017) mitigates this by broadening the training distribution and has shown strong empirical success in robotic control, sim-to-real transfer, and visual generalization (Sen et al., 2025; Chen et al., 2021; Yue et al., 2019). However, DR samples levels from a fixed parameter distribution, which often yields levels that are either too easy or too hard for the current agent, wasting interaction budget and hindering generalization.

Unsupervised Environment Design (UED) (Dennis et al., 2020) addresses this limitation by adapting the training distribution to the learning state of the agent. The core challenge in UED is to quantify the training value of a level $l$ for the current policy $\pi_{\theta_{old}}$—a quantity we denote as the level score $S(l)$—and use it to select, replay, edit, or generate more useful training levels. This score governs the induced curriculum and remains a key design choice across UED systems. Representative paradigms include adversarial generation in PAIRED (Dennis et al., 2020), replay-based prioritization in PLR and robust PLR (Jiang et al., 2021b;a), evolutionary editing in ACCEL (Parker-Holder et al., 2022), and marginal-benefit estimation in MBeDED (Li et al., 2025). Recent studies refine level scoring through learnability-aware regret design, trajectory-level regret approximation, active level selection, distribution

grounding, and optimization-theoretic formulations (Beukman et al., 2024; Cho et al., 2025; Jang et al., 2024; Jiang et al., 2022; Monette et al., 2025). Other methods expand curriculum coverage by adding diversity, novelty, or state-action coverage signals (Li et al., 2023; Xiang et al., 2024; Teoh et al., 2024; Garcin et al., 2024). A complementary direction broadens the UED design space through diffusion-based generation, hierarchical teacher-student interaction, partial environment generation, and joint task-level design (Chung et al., 2024; Li et al., 2026; Mead et al., 2026; Furelos-Blanco et al., 2026). Together, these studies show that UED progress depends not only on how levels are generated, replayed, or edited, but also on how reliably their training value is measured.

PACE follows this line of research but adopts a distinct perspective on level evaluation. Existing methods construct $S(l)$ using proxies such as regret estimates, trajectory-level statistics (e.g., GAE, TD error), or comparisons of policy performance before and after updates—often requiring additional rollouts, value-function estimation, or auxiliary models. In contrast, PACE computes the provisional policy update $\Delta J(\theta)$ induced by $l$ under $\pi_{\theta_{old}}$, and uses the resulting parameter change $\|\Delta\theta\|$ as an intrinsic signal for $S(l)$. This design avoids extra interaction or rollback evaluation, directly ties level scoring to the local optimization progress, and provides a simple, low-variance scoring module compatible with existing UED pipelines.

## 6. Conclusion

We present Parameter Change Environment Design (PACE), a simple and effective framework for Unsupervised Environment Design that evaluates environments through policy parameter change. Specifically, PACE uses an intrinsic and low-variance signal that reflects agent realized learning progress, without relying on value estimation, regret approximation, or Monte Carlo return comparison. Theoretically , A first-order theoretical analysis motivates a direct relationship between environment-induced policy updates and improvement in the policy optimization objective. This analysis provides a principled foundation for using policy parameter change as an environment value signal. Empirically results on MiniGrid and large-scale Craftax benchmarks show that PACE consistently improves generalization and remains robust under long-horizon and non-stationary training dynamics. Overall, PACE offers a practical and principled alternative to existing UED evaluation criteria by grounding environment value in realized learning progress. More broadly, it reframes environment evaluation around realized optimization progress, and highlights policy parameter change as a reliable signal for guiding environment selection in reinforcement learning.

## Acknowledgements

This work was supported by the National Natural Science Foundation of China (Grant No. 62402507) and the Innovation Capability Support Program of Shaanxi Province (Grant No. 2024ZC-KJXX-072).

## Impact Statement

This paper presents work whose goal is to advance the field of Machine Learning. Specifically, we propose PACE, a methodological improvement for Unsupervised Environment Design (UED) in reinforcement learning. By using policy parameter change to estimate the training value of environments, PACE improves out-of-distribution generalization. Our experiments are foundational and conducted entirely in simulated benchmarks (MiniGrid and Craftax). There are many potential societal consequences of our work, none which we feel must be specifically highlighted here.

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

# Supplementary Material for
# "PACE: Parameter Change for Unsupervised Environment Design"

This appendix is organized into three parts: extended related work, additional analysis of the PACE score, and supplementary experimental results. The table of contents below summarizes the structure of the **Appendix**.

## Appendix Contents

## A. Extended Related Work

Generalization remains a central challenge in deep RL, where policies that perform well on training environments often fail under shifts in dynamics, observations, layouts, or task parameters (Ma et al., 2023; Queeney et al., 2025; Huang et al., 2025). Domain Randomization (DR) addresses this issue by broadening the training distribution through randomized environment parameters and has shown strong empirical value in robotic control, sim-to-real transfer, and visual generalization (Tobin et al., 2017; Huber et al., 2024). However, DR usually relies on a predefined sampling distribution and does not explicitly evaluate whether a sampled level matches the current capability of the agent. UED offers a more adaptive alternative by adjusting the training distribution according to the learning state of the agent (Dennis et al., 2020). Rather than sampling levels from a fixed distribution, UED constructs curricula by selecting, replaying, editing, or generating levels that provide useful learning signals. Representative paradigms include adversarial level generation in PAIRED (Dennis et al., 2020), replay-based curricula in PLR and robust PLR (Jiang et al., 2021b;a), evolutionary level editing in ACCEL (Parker-Holder et al., 2022), and marginal-benefit-based level evaluation in MBeDED (Li et al., 2025); detailed discussions of these methods appear in Section 2.

A core question in UED is how to define the score $S$ that measures the training value of a level for the current agent (Jiang et al., 2021b; Cho et al., 2025; Jang et al., 2024; Monette et al., 2025). This score influences which levels enter the level buffer $\Lambda$, which levels are replayed, and which levels receive further adaptation. Early replay-based methods typically prioritize levels with trajectory-level proxy signals, such as GAE, TD error, or returns (Jiang et al., 2021b;a). While effective, these signals only provide indirect estimates of learning progress and can suffer from value-estimation bias or high variance. Recent methods therefore refine level scoring by moving beyond raw difficulty. For instance, ReMiDi (Beukman et al., 2024) addresses regret stagnation in regret-based environment design by showing that consistently high-regret levels can stall learning. It therefore prioritizes levels that remain learnable, i.e., levels that offer room for improvement without being hopeless or already solved. Building on a richer view of agent behavior, TRACED (Cho et al., 2025) incorporates trajectory-level statistics, such as transition prediction error, to better approximate regret and account for learning transfer across tasks. ActivePLR (Jang et al., 2024) takes a different route by casting replay-based UED as an active selection problem, directly searching for levels that are expected to be informative for the current agent. Beyond scoring itself, SAMPLR addresses curriculum-induced covariate shift by grounding policy updates to the true environment distribution

through fictitious transitions, allowing curricula to remain challenging without biasing the learned policy (Jiang et al., 2022). From an optimization perspective, NCC (Monette et al., 2025) reformulates UED as a nonconvex-strongly-concave game with entropy regularization and provides convergence guarantees for a two-timescale stochastic gradient descent-ascent algorithm. These studies reflect a broader shift from maximizing difficulty toward designing level scores that better align with realized learning progress.

Another line of work improves UED by encouraging diversity, novelty, and broader coverage in the training distribution (Li et al., 2023; Xiang et al., 2024; Teoh et al., 2024; Garcin et al., 2024). Rather than focusing only on level difficulty, these methods combine learning-potential estimates with explicit measures of distributional spread or state-action coverage. DIPLR, for example, measures level similarity through Wasserstein distance between occupancy distributions, and uses this signal to select or generate levels that are both challenging and distinct (Li et al., 2023). Moving from occupancy-level diversity to novelty in state-action space, GENIE (Xiang et al., 2024) fits a Gaussian mixture model to ground-truth state-action pairs and scores candidate levels by their negative log-likelihood. CENIE (Teoh et al., 2024) follows a similar motivation but makes the novelty estimate agent-centric: it models historical state-action coverage from the agent and assigns higher scores to levels that deviate from past experience. When integrated with PLR or ACCEL, these novelty signals encourage curricula that expand state-action coverage rather than repeatedly revisiting similar challenges. DRED (Garcin et al., 2024) takes a more theoretical route by linking zero-shot generalization to minimizing mutual information between the policy and training levels. It further combines adaptive sampling with a variational autoencoder that approximates the target CMDP distribution, enabling additional level generation while controlling distributional shift. Together, these methods differ in how they quantify diversity or novelty, but they share the same goal of preventing adaptive curricula from collapsing to a narrow set of difficult yet redundant levels.

Recent UED studies also expand how environments are generated and how teachers, students, tasks, and levels interact (Chung et al., 2024; Li et al., 2026; Mead et al., 2026; Furelos-Blanco et al., 2026). A prominent direction brings generative modeling into UED. For example, ADD (Chung et al., 2024) combines learning-based and replay-based UED with a diffusion generator, where a distributional environment critic provides a differentiable regret estimate to guide environment generation. This formulation further leads to Soft UED, which introduces entropy regularization and yields a closed-form optimal environment distribution with zero duality gap. With regret-guided diffusion sampling, the generated curricula remain both challenging and diverse. Beyond the generator itself, recent work also studies how teachers and students should interact during environment design. In SHED (Li et al., 2026), UED is formulated as a hierarchical MDP in which a teacher generates training environments according to student performance on evaluation levels. To lower the interaction cost between teacher and student, a conditional diffusion model serves as a world model, synthesizing student state transitions and mixing them with real experience for teacher learning. Other approaches focus on improving credit assignment for environment generation. Instead of generating a full environment in one step, DEGen (Mead et al., 2026) generates only the part of the environment observed by the student, which provides denser teacher rewards and reduces feedback noise. It also introduces Maximised Negative Advantage (MNA), a GAE-based regret approximation that identifies challenging levels while penalizing unsolvable ones. The design space further extends beyond levels: ATLAS (Furelos-Blanco et al., 2026) jointly designs tasks and levels, specifies tasks as reward machines, and uses regret-based prioritization with structure-aware mutations to construct curricula over solvable yet challenging task-level pairs. These methods broaden the UED design space, but their effectiveness still depends on reliable evaluation signals that identify which generated levels are useful for the current agent.

Taken together, prior work shows that UED progress depends not only on how levels are generated, replayed, or edited, but also on how their training value is measured (Jiang et al., 2021a;b; Beukman et al., 2024; Cho et al., 2025; Jang et al., 2024; Li et al., 2025). Existing methods estimate level usefulness through regret, return, trajectory-level statistics, novelty, diversity, teacher objectives, or explicit comparisons before and after policy updates. These signals provide effective curricula in many settings, but they often require additional rollout information, value-function-based estimation, generative modeling, or carefully designed proxy objectives. PACE follows this scoring-centric view but evaluates level value from the perspective of policy optimization. For a fixed level $l$, the agent collects trajectory $\tau$ under $\pi_{\theta_{\mathrm{old}}}$ and computes the provisional update induced by this level. PACE uses the resulting parameter change as an intrinsic signal for the score $S$, so the score reflects how strongly $l$ moves the current policy. This design directly couples level evaluation with the optimization process, requires no extra environment interaction beyond the trajectories already collected for policy updates, avoids rollback evaluation, and provides a simple scoring module that can complement replay-based, editing-based, and generation-based UED systems.

# B. Extended Analysis of the PACE Score

## B.1. Second-order Approximation and Local Validity

We further analyze the omitted second-order term and clarify the local validity of this approximation. Let $\Delta\theta = \theta_{\text{new}} - \theta_{\text{old}}$ denote the parameter change induced by level $l$. Extending Eq. 7 to the second order gives

$$\Delta J(l) \approx \nabla_\theta J(\theta_{\text{old}}, l)^\top \Delta\theta + \frac{1}{2}\Delta\theta^\top H_l \Delta\theta. \tag{15}$$

where $H_l = \nabla_\theta^2 J(\theta_{\text{old}}, l)$ denotes the local Hessian for level $l$. Under the gradient-aligned update assumption in Eq. 9, we have $\nabla_\theta J(\theta_{\text{old}}, l) = \frac{1}{\alpha}\Delta\theta$. Therefore, the first-order term becomes

$$\begin{aligned}
\nabla_\theta J(\theta_{\text{old}}, l)^\top \Delta\theta &= \left(\frac{1}{\alpha}\Delta\theta\right)^\top \Delta\theta \\
&= \frac{1}{\alpha}\Delta\theta^\top \Delta\theta \\
&= \frac{1}{\alpha}\|\Delta\theta\|_2^2 \\
&= S(l).
\end{aligned} \tag{16}$$

where the last equality follows from the PACE score definition in Eq. 11. Substituting this result into the second-order expansion yields

$$\Delta J(l) \approx S(l) + \frac{1}{2}\Delta\theta^\top H_l \Delta\theta. \tag{17}$$

**Local validity.** The derivation relies on a local small-update regime. In our implementation, the PACE score is computed from a provisional policy update with a small learning rate, and this provisional update is discarded after scoring. PPO clipping further encourages local policy changes during training. These mechanisms keep the scoring procedure close to the local regime in practice, although they do not impose a strict bound on the parameter change $\Delta\theta$.

**Approximation error and limitations.** Dropping the second-order term introduces an approximation error, which constitutes a limitation of the first-order derivation. This error depends on the local curvature encoded by the Hessian $H_l$. If the local curvature is bounded as $\|H_l\|_2 \leq M_l$ around $\theta_{\text{old}}$, then

$$\left|\frac{1}{2}\Delta\theta^\top H_l \Delta\theta\right| \leq \frac{1}{2}M_l\|\Delta\theta\|_2^2. \tag{18}$$

Therefore, the omitted term scales with $\|\Delta\theta\|_2^2$. Under the single-step update assumption, $\|\Delta\theta\|_2$ is proportional to the step size $\alpha$. Consequently, the first-order PACE score in Eq. (6) scales as $O(\alpha)$, whereas the omitted second-order term scales as $O(\alpha^2)$ when the gradient norm and local curvature remain bounded. In the small-update regime, the first-order term therefore dominates the approximation.

This analysis also clarifies when the approximation can become less accurate. If the update is large, or if different levels induce substantially different local curvature, the second-order term may perturb the ideal ranking of levels. We therefore interpret the first-order relationship as theoretical motivation for PACE rather than an exact characterization of deep RL optimization dynamics.

## B.2. Robustness under Stochastic Gradients

The derivation in Section 3.1 assumes a deterministic gradient update. In practice, the policy gradient estimate is stochastic due to environment randomness and finite-sample trajectory collection. We extend the analysis to account for this noise and characterize its effect on the PACE score.

Consider a stochastic gradient estimate on level $l$:

$$\hat{g}_l = \nabla_\theta J(\pi_{\theta_{\text{old}}}, l) + \epsilon_l. \tag{19}$$

where $\epsilon_l$ denotes zero-mean estimation noise and $\mathbb{E}[\|\epsilon_l\|_2^2] = \sigma_l^2$. The corresponding provisional parameter update is $\Delta\theta_l = \alpha\hat{g}_l$. Then,

$$\mathbb{E}_{\epsilon_l}\left[\|\Delta\theta_l\|_2^2\right] = \alpha^2\|\nabla_\theta J(\pi_{\theta_{\text{old}}}, l)\|_2^2 + \alpha^2\sigma_l^2. \tag{20}$$

where the cross term vanishes because $\mathbb{E}[\epsilon_l] = 0$.

Under the same first-order approximation used in Section 3.1, the expected objective improvement satisfies

$$\mathbb{E}_{\epsilon_l}[\Delta J(\theta)] \approx \nabla_\theta J(\pi_{\theta_{\text{old}}}, l)^\top \mathbb{E}_{\epsilon_l}[\Delta\theta_l] = \alpha\|\nabla_\theta J(\pi_{\theta_{\text{old}}}, l)\|_2^2. \tag{21}$$

Substituting this relation into Eq. (20) gives

$$\mathbb{E}_{\epsilon_l}\left[\|\Delta\theta_l\|_2^2\right] \approx \alpha\,\mathbb{E}_{\epsilon_l}[\Delta J(\theta)] + \alpha^2\sigma_l^2. \tag{22}$$

Eq. (22) shows that the expected parameter-change score contains two components: a true learning-progress term and a residual noise term induced by stochastic gradient estimation. Therefore, PACE does not require the parameter-change norm to equal the objective improvement exactly. Instead, the score serves as a noisy proxy whose expectation remains aligned with the objective improvement up to the residual term $\alpha^2\sigma_l^2$.

Moreover, standard mini-batch gradient estimation gives $\sigma_l^2 = O(1/B)$, where $B$ denotes the number of sampled transitions. Thus, increasing the trajectory batch size reduces the residual noise term. This analysis clarifies that environment randomness and data noise can affect the PACE score, but their contribution enters as a variance-controlled residual rather than as a systematic bias in the learning-progress term.

Beyond the variance analysis above, we also consider the practical validity of the gradient-aligned assumption under finite-sample estimation. Ilyas et al. (2018) show that finite-sample policy gradient estimates in deep RL can exhibit low cosine similarity with the true gradient, raising the question of whether the assumption underlying Eq. 10 remains meaningful in practice. PACE mitigates this concern through two design choices. First, the scoring phase uses only a single provisional update on trajectories collected from level $l$, and the resulting parameter change is discarded after computing $S(l)$. This design directly matches the single-step update considered in Section 3.1 and keeps score computation separate from subsequent policy training. Second, PACE selects levels by relative score ranking (Eq. 12) rather than absolute score values. Thus, the scoring mechanism does not require each finite-sample gradient estimate to match the true gradient accurately; it only requires the induced scores to preserve a useful ordering among candidate levels, which is a weaker condition and aligns with the replay-based selection objective.

## B.3. Comparison with Loss-Based Scoring Signals

The PACE score is related to, but fundamentally different from, loss-magnitude-based scores in the PLR family. Under the idealized single-step gradient update, the PACE score for a level $l$ expands as

$$S(l) = \alpha\|g_l\|_2^2, \quad g_l = \nabla_\theta J(\theta, l). \tag{23}$$

With a GAE-based policy-gradient estimator, $g_l \approx \frac{1}{T}\sum_t \nabla_\theta \log \pi_\theta(a_t|s_t)\,\hat{A}_t$, which gives

$$S(l) \approx \frac{\alpha}{T^2}\,\hat{\mathbf{A}}^\top K_l\,\hat{\mathbf{A}}. \tag{24}$$

where $K_l$ is the Gram matrix formed by the policy-score vectors $\{\nabla_\theta \log \pi_\theta(a_t|s_t)\}_t$ and $\hat{\mathbf{A}} = (\hat{A}_1, \ldots, \hat{A}_T)^\top$ collects per-step advantage estimates. This decomposition reveals that the PACE score depends jointly on the magnitude of advantage estimates and the alignment among per-step gradient directions.

In contrast, scoring functions in the PLR family aggregate per-step scalar signals without accounting for gradient geometry. PLR (Jiang et al., 2021b) uses L1 value loss (GAE magnitude), $\frac{1}{T}\sum_t |\hat{A}_t|$, which averages absolute per-step GAE values. Robust PLR (Jiang et al., 2021a) argues that while L1 value loss aids rapid value-function training, it lacks regret guarantees and can bias long-term training toward high-variance policies. Robust PLR therefore replaces L1 value loss with Positive Value Loss (PVL), $\frac{1}{T}\sum_t \max(\hat{A}_t, 0)$, which retains only positive advantage terms as a biased, optimistic estimate of regret. ACCEL (Parker-Holder et al., 2022) adopts the same PVL scoring.

All these scoring functions aggregate per-step scalar error signals and do not capture interactions among gradient directions. A level with large per-step errors (large $\sum_t |\hat{A}_t|$) can still induce a small parameter update when per-step gradient contributions

cancel, as $\hat{\mathbf{A}}^\top K_l\, \hat{\mathbf{A}}$ can be much smaller than $\|\hat{\mathbf{A}}\|_1^2$. Conversely, a level with moderate per-step errors can induce a larger update when gradient contributions align. By directly measuring the induced change in the policy parameter space, PACE inherently captures these alignment and cancellation effects.

## B.4. Complexity and Stability Analysis

We compare PACE with MBeDED (Li et al., 2025) from two perspectives: computational complexity and evaluation stability. We focus on MBeDED because it directly targets the objective increment $\Delta J$ as the environment value signal, making it the closest existing UED method to PACE in motivation.

**Evaluation Complexity.** MBeDED evaluates environment value through a dual evaluation procedure. After obtaining updated parameters $\theta_{\mathrm{new}}$, it performs $k$ additional rollout episodes of horizon $T$ to estimate the marginal improvement. Let $P$ denote the number of policy parameters. Since each forward pass has complexity $O(P)$, the additional evaluation cost of MBeDED is

$$C_{\mathrm{MBeDED}} \approx O(k \cdot T \cdot P). \tag{25}$$

In contrast, PACE evaluates a level by computing the squared norm of the policy parameter difference. This operation involves only tensor arithmetic in memory and requires no further environment interaction:

$$C_{\mathrm{PACE}} \approx O(P). \tag{26}$$

In typical reinforcement learning settings, $k \cdot T$ ranges from $10^3$ to $10^4$. As a result, PACE reduces the evaluation cost by several orders of magnitude and incurs no additional sampling cost beyond the trajectory collected for the level scoring step.

**Evaluation Stability.** The MBeDED estimate $\Delta\hat{J}_{\mathrm{MBeDED}}$ is computed as the difference between two Monte Carlo return estimates. Its variance therefore satisfies

$$\mathrm{Var}(\Delta\hat{J}_{\mathrm{MBeDED}}) \approx \mathrm{Var}(V_{\mathrm{new}}) + \mathrm{Var}(V_{\mathrm{old}}). \tag{27}$$

In sparse-reward or highly stochastic environments, this compounded variance can dominate the true improvement signal, which often results in noisy and oscillatory environment evaluation.

By contrast, the PACE score $\|\Delta\theta\|_2^2$ provides a deterministic measurement of the policy parameter update induced by a level. Although the update depends on stochastic training batches, PACE avoids the additional sampling noise introduced by explicit policy re-evaluation. This tight coupling between the evaluation signal and the optimization step yields a more consistent estimate of learning progress, which substantially improves curriculum stability and convergence behavior.

We also conduct a preliminary comparison with MBeDED (Li et al., 2025) on the same MiniGrid benchmark. Figure 10 presents the aggregate OOD performance: the left panel reproduces our results from Figure 4, and the right panel reproduces Figure 6 from Li et al. (2025). PACE achieves an IQM in $[0.8, 1.0]$ and an Optimality Gap in $[0.0, 0.2]$, whereas MBeDED reports an IQM in $[0.6, 0.8]$ and an Optimality Gap in $[0.2, 0.4]$. These preliminary results suggest that PACE remains competitive in aggregate OOD performance. A more thorough and controlled comparison is left to future work.

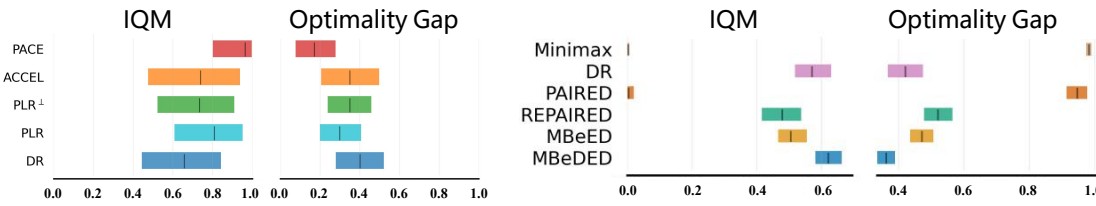

*Figure 10.* Preliminary comparison with MBeDED on MiniGrid. The left panel shows our aggregate OOD results from Figure 4, and the right panel reproduces Figure 6 from Li et al. (2025). Higher IQM and lower Optimality Gap indicate better performance.

# C. Additional Experimental Results

This appendix provides full per-level results on MiniGrid (Appendix C.1) and Craftax (Appendix C.2), per-level score-stability statistics that complement Section 4.3 (Appendix C.3), and the hyperparameters used in all experiments (Appendix C.4). Table 2 reports the total number of environment interactions for the same student PPO budget across methods.

*Table 2.* Total number of environment interactions for a given number of student PPO updates.

| Environment | PPO Updates | PACE | ACCEL | PLR$^\perp$ | PLR | DR |
|---|---|---|---|---|---|---|
| MiniGrid | 30 000 | 245.76M | 245.76M | 245.76M | 245.76M | 245.76M |
| Craftax | 255 | 1.07B | 1.07B | 1.07B | 1.07B | 1.07B |

## C.1. MiniGrid Evaluation Results

We report extended evaluation results on a set of 12 held-out MiniGrid levels commonly used in the UED literature (Parker-Holder et al., 2022), illustrated in Figure 11. These levels span a range of structural complexities and are designed to test zero-shot generalization to environments that differ substantially from those encountered during training. All methods are evaluated after the same number of student PPO updates, following the protocol used in ACCEL and PLR.

Figure 12 reports per-level solved rates with mean and standard deviation across 10 training seeds, and Table 3 provides the corresponding numerical results. Table 4 further summarizes aggregate performance via the Interquartile Mean (IQM) and Optimality Gap of solved rates.

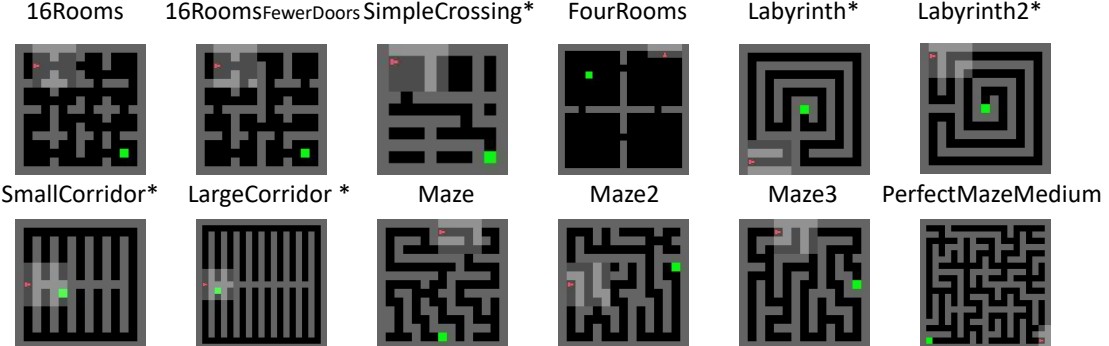

*Figure 11.* MiniGrid zero-shot evaluation levels used for transfer evaluation.

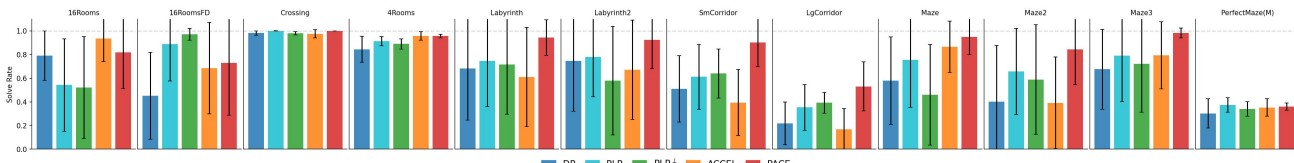

*Figure 12.* Zero-shot transfer results in out-of-distribution mazes. Agents are evaluated for 100 episodes on human-designed mazes. Plots show mean and standard deviation for each environment, across 10 runs.

## C.2. Craftax Evaluation Results

We report evaluation results on the Craftax benchmark following the Craftax-1B protocol. After training for a fixed interaction budget, we evaluate the final policy checkpoints on a held-out set of 20 Craftax levels. Unlike MiniGrid, performance in Craftax is measured by episodic reward rather than solved rate. All reported results therefore use reward-based metrics, consistent with the evaluation setup described in the main paper.

Figure 13 reports per-level episodic reward with mean and standard deviation across 10 training seeds, and Table 5 provides the corresponding numerical results. Table 6 further summarizes aggregate performance via the Interquartile Mean (IQM) and Optimality Gap of episodic reward.

*Table 3.* Zero-shot solved rate on 12 held-out MiniGrid levels. Each entry reports the mean $\pm$ standard deviation over 10 training seeds. Bold values indicate the highest solved rate for each level.

| Level | DR | PLR | PLR$^\perp$ | ACCEL | PACE |
|---|---|---|---|---|---|
| 16Rooms | $0.791 \pm 0.209$ | $0.542 \pm 0.391$ | $0.520 \pm 0.431$ | $\mathbf{0.934} \pm 0.195$ | $0.819 \pm 0.305$ |
| 16RoomsFD | $0.451 \pm 0.368$ | $0.887 \pm 0.310$ | $\mathbf{0.972} \pm 0.050$ | $0.685 \pm 0.386$ | $0.728 \pm 0.441$ |
| Crossing | $0.981 \pm 0.018$ | $1.000 \pm 0.001$ | $0.980 \pm 0.014$ | $0.975 \pm 0.035$ | $\mathbf{1.000} \pm 0.000$ |
| 4Rooms | $0.844 \pm 0.110$ | $0.912 \pm 0.038$ | $0.889 \pm 0.042$ | $0.956 \pm 0.036$ | $\mathbf{0.957} \pm 0.015$ |
| SmCorridor | $0.510 \pm 0.281$ | $0.612 \pm 0.274$ | $0.640 \pm 0.207$ | $0.394 \pm 0.280$ | $\mathbf{0.901} \pm 0.201$ |
| LgCorridor | $0.218 \pm 0.181$ | $0.354 \pm 0.193$ | $0.393 \pm 0.088$ | $0.168 \pm 0.174$ | $\mathbf{0.530} \pm 0.207$ |
| Labyrinth | $0.682 \pm 0.435$ | $0.747 \pm 0.387$ | $0.716 \pm 0.420$ | $0.610 \pm 0.420$ | $\mathbf{0.943} \pm 0.150$ |
| Labyrinth2 | $0.747 \pm 0.425$ | $0.781 \pm 0.337$ | $0.580 \pm 0.459$ | $0.671 \pm 0.419$ | $\mathbf{0.923} \pm 0.240$ |
| Maze | $0.580 \pm 0.370$ | $0.754 \pm 0.399$ | $0.460 \pm 0.425$ | $0.865 \pm 0.216$ | $\mathbf{0.950} \pm 0.152$ |
| Maze2 | $0.403 \pm 0.473$ | $0.658 \pm 0.364$ | $0.588 \pm 0.463$ | $0.392 \pm 0.387$ | $\mathbf{0.844} \pm 0.299$ |
| Maze3 | $0.675 \pm 0.338$ | $0.791 \pm 0.389$ | $0.721 \pm 0.408$ | $0.794 \pm 0.283$ | $\mathbf{0.983} \pm 0.042$ |
| PerfectMaze(M) | $0.302 \pm 0.124$ | $\mathbf{0.373} \pm 0.061$ | $0.339 \pm 0.061$ | $0.352 \pm 0.074$ | $0.359 \pm 0.030$ |
| **Mean** | $0.599 \pm 0.230$ | $0.701 \pm 0.202$ | $0.650 \pm 0.214$ | $0.650 \pm 0.270$ | $\mathbf{0.828} \pm \mathbf{0.198}$ |

*Table 4.* Aggregate performance across 12 held-out MiniGrid levels. We report the Interquartile Mean (IQM) and Optimality Gap of solved rates, computed using the `rliable` library. Values in brackets denote 95% confidence intervals. Bold values indicate the best performance for each metric.

| Method | IQM $\uparrow$ | Optimality Gap $\downarrow$ |
|---|---|---|
| DR | 0.657 [0.444, 0.842] | 0.401 [0.280, 0.521] |
| PLR | 0.808 [0.607, 0.950] | 0.299 [0.200, 0.407] |
| PLR$^\perp$ | 0.733 [0.521, 0.908] | 0.350 [0.239, 0.458] |
| ACCEL | 0.739 [0.475, 0.937] | 0.350 [0.206, 0.498] |
| PACE | **0.964** [0.800, 0.997] | **0.172** [0.077, 0.278] |

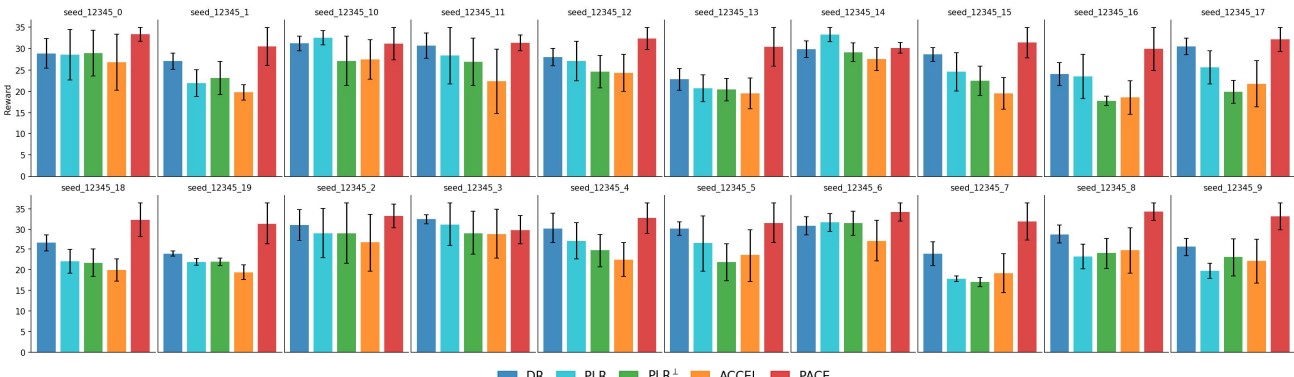

*Figure 13.* Evaluation reward on 20 held-out Craftax levels. Bars report the mean episodic reward and error bars indicate one standard deviation across training seeds. All methods are evaluated using the same final policy checkpoints.

## C.3. Score Stability Results

This section provides the detailed coefficient of variation (CV) computation referenced in Section 4.3, followed by per-pair stability results across fixed policy-level pairs.

For each pair $(\pi_{\theta_i}, l_j)$, we compute the mean score

$$\mu_{i,j} = \frac{1}{R} \sum_{r=1}^{R} \hat{S}^{(r)}(l_j; \pi_{\theta_i}), \tag{28}$$

the standard deviation

$$\sigma_{i,j} = \text{Std}_r \left[ \hat{S}^{(r)}(l_j; \pi_{\theta_i}) \right], \tag{29}$$

*Table 5.* Per-level evaluation reward on 20 Craftax levels. Each entry reports the mean ± standard deviation of episodic reward over 10 training seeds. The Mean row reports the average and standard deviation across levels.

| Level | DR | PLR | PLR$^\perp$ | ACCEL | PACE |
|---|---|---|---|---|---|
| 0 | 28.871 ± 3.433 | 28.545 ± 5.869 | 28.950 ± 5.355 | 26.828 ± 6.521 | **33.315** ± 1.629 |
| 1 | 24.893 ± 1.718 | 20.184 ± 2.909 | 21.277 ± 3.524 | 18.193 ± 1.643 | **28.072** ± 4.066 |
| 2 | 30.349 ± 3.676 | 28.403 ± 5.957 | 28.402 ± 7.290 | 26.073 ± 6.845 | **32.559** ± 2.824 |
| 3 | 32.687 ± 1.120 | **31.375** ± 5.362 | 29.265 ± 5.382 | 29.050 ± 6.128 | 30.016 ± 3.608 |
| 4 | 25.841 ± 3.165 | 23.198 ± 3.848 | 21.143 ± 3.427 | 19.205 ± 3.488 | **28.012** ± 3.201 |
| 5 | 26.968 ± 1.472 | 23.602 ± 6.079 | 19.450 ± 4.023 | 21.009 ± 5.712 | **28.111** ± 4.446 |
| 6 | 31.158 ± 2.214 | 32.004 ± 2.210 | 31.798 ± 2.922 | 27.393 ± 5.102 | **34.553** ± 2.275 |
| 7 | 22.352 ± 2.704 | 16.676 ± 0.631 | 15.938 ± 1.034 | 17.936 ± 4.445 | **29.909** ± 4.267 |
| 8 | 28.412 ± 2.234 | 22.949 ± 2.936 | 23.777 ± 3.729 | 24.449 ± 5.563 | **33.932** ± 2.117 |
| 9 | 24.446 ± 2.094 | 18.862 ± 1.704 | 22.103 ± 4.395 | 21.192 ± 5.198 | **31.729** ± 3.149 |
| 10 | 31.349 ± 1.725 | **32.661** ± 1.679 | 27.255 ± 5.785 | 27.566 ± 4.607 | 31.293 ± 3.809 |
| 11 | 33.722 ± 3.215 | 31.144 ± 7.251 | 29.566 ± 6.015 | 24.571 ± 8.218 | **34.439** ± 2.051 |
| 12 | 27.643 ± 1.979 | 26.768 ± 4.525 | 24.309 ± 3.709 | 23.997 ± 4.269 | **31.926** ± 2.560 |
| 13 | 23.464 ± 2.597 | 21.359 ± 3.222 | 21.012 ± 2.678 | 20.068 ± 3.707 | **31.244** ± 4.665 |
| 14 | 30.899 ± 2.039 | **34.453** ± 1.724 | 30.154 ± 2.231 | 28.534 ± 2.783 | 31.233 ± 1.296 |
| 15 | 29.230 ± 1.643 | 25.090 ± 4.537 | 22.961 ± 3.464 | 19.952 ± 3.718 | **32.044** ± 3.608 |
| 16 | 24.151 ± 2.652 | 23.628 ± 5.150 | 17.880 ± 1.063 | 18.694 ± 3.910 | **30.034** ± 5.059 |
| 17 | 32.087 ± 2.053 | 26.955 ± 4.072 | 20.935 ± 2.790 | 22.915 ± 5.686 | **33.784** ± 2.969 |
| 18 | 26.609 ± 2.081 | 22.049 ± 2.909 | 21.706 ± 3.282 | 19.940 ± 2.698 | **32.380** ± 4.131 |
| 19 | 19.323 ± 0.525 | 17.701 ± 0.658 | 17.715 ± 0.762 | 15.664 ± 1.438 | **25.368** ± 4.137 |
| **Mean** | 27.723 ± 3.839 | 25.380 ± 5.233 | 23.780 ± 4.688 | 22.661 ± 4.005 | **31.198 ± 2.439** |

*Table 6.* Aggregate performance across 20 Craftax evaluation levels. We report the Interquartile Mean (IQM) and Optimality Gap of episodic reward, computed using the `rliable` library. Values in brackets denote 95% confidence intervals. Bold values indicate the best performance for each metric.

| Method | IQM ↑ | Optimality Gap ↓ |
|---|---|---|
| DR | 0.603 [0.530, 0.675] | 0.407 [0.343, 0.470] |
| PLR | 0.484 [0.372, 0.615] | 0.495 [0.409, 0.580] |
| PLR$^\perp$ | 0.409 [0.320, 0.520] | 0.556 [0.479, 0.633] |
| ACCEL | 0.364 [0.283, 0.458] | 0.599 [0.534, 0.664] |
| PACE | **0.750** [0.706, 0.788] | **0.268** [0.232, 0.305] |

and the corresponding coefficient of variation, together with its average over all policy-level pairs:

$$\mathrm{CV}_{i,j} = \frac{\sigma_{i,j}}{\mu_{i,j}}, \qquad \overline{\mathrm{CV}} = \frac{1}{NM} \sum_{i=1}^{N} \sum_{j=1}^{M} \mathrm{CV}_{i,j}, \tag{30}$$

where $N$, $M$, and $R$ denote the number of checkpoints, fixed levels, and repetitions per pair, respectively.

Building on this metric, we report the per-pair CV of score estimates under fixed policy-level pairs. Table 7 reports the results on five held-out MiniGrid levels, and Table 8 reports the results on five randomly generated Craftax levels. Each entry is the CV computed from $R$ independently sampled trajectories on the same $(\pi_{\theta_i}, l_j)$ pair, averaged over $N$ checkpoints. PACE consistently produces the lowest score-estimation variability across levels on both benchmarks, with CV reaching zero on multiple MiniGrid levels.

## C.4. Hyperparameters

The majority of hyperparameters are inherited from prior work on unsupervised environment design (Dennis et al., 2020; Jiang et al., 2021b;a; Parker-Holder et al., 2022), with only minor modifications. For both MiniGrid and Craftax, we conduct a grid search over the level replay buffer size $\{500, 1000, 2000, 4000, 6000, 8000\}$, the replay probability $\{0.3, 0.5, 0.6, 0.7, 0.8\}$, and the temperature parameter $\beta \in \{0.3, 0.5, 0.6, 0.7, 0.8, 0.9, 1.0\}$. Hyperparameters are selected based on performance on held-out validation levels.

For MiniGrid, we follow the evaluation protocol used in ACCEL and PLR, and adopt the same set of 12 human-designed

*Table 7.* Per-level CV of score estimates on five held-out MiniGrid levels. Lower is better. The lowest value in each row is shown in bold.

| Level | PLR | PLR$^{\perp}$ | ACCEL | PACE |
|---|---|---|---|---|
| 6Rooms | 0.0056 | 0.0041 | 0.0055 | **0.0000** |
| Labyrinth | 0.0109 | 0.0015 | 0.0101 | **0.0000** |
| Labyrinth2 | 0.0210 | 0.0057 | 0.0219 | **0.0000** |
| Maze | **0.0000** | 0.0149 | 0.0114 | **0.0000** |
| Maze2 | 0.0018 | **0.0011** | 0.0188 | 0.0051 |
| Mean CV | 0.0079 | 0.0055 | 0.0135 | **0.0010** |

*Table 8.* Per-level CV of score estimates on five randomly generated Craftax levels. Lower is better. The lowest value in each row is shown in bold.

| Level | PLR | PLR$^{\perp}$ | ACCEL | PACE |
|---|---|---|---|---|
| level 1 | **0.3658** | 0.6072 | 0.3804 | 0.5920 |
| level 2 | 0.4813 | 0.5969 | 0.7016 | **0.4365** |
| level 3 | 0.7678 | 0.7801 | 0.6270 | **0.5373** |
| level 4 | 0.6539 | 0.7944 | 0.5439 | **0.4994** |
| level 5 | 0.7838 | 0.6842 | 0.8756 | **0.6406** |
| Mean CV | 0.6105 | 0.6926 | 0.6257 | **0.5412** |

levels as validation environments. For Craftax, we follow the experimental setup from (Matthews et al., 2024), and evaluate on the Craftax-1B Challenge using the Craftax-Symbolic environment. We randomly generate 20 evaluation levels and use them as validation levels for hyperparameter selection.

The final hyperparameters used in all experiments are reported in Table 9.

*Table 9.* Hyperparameters used in all experiments.

| Parameter | MiniGrid | Craftax |
|---|---|---|
| **PPO** | | |
| Learning rate | $1 \times 10^{-4}$ | $3 \times 10^{-4}$ ($2 \times 10^{-4}$ DR) |
| Max gradient norm | 0.5 | 1.0 |
| Number of PPO updates | 30,000 | 255 |
| Rollout length | 256 | – |
| Inner rollout length | – | 64 |
| Outer rollout length | – | 64 |
| Number of environments | 32 | 1024 |
| Minibatches per update | 1 | 2 |
| Discount factor $\gamma$ | 0.995 | 0.995 |
| PPO epochs | 5 | 5 |
| Clip range | 0.2 | 0.2 |
| GAE parameter $\lambda$ | 0.98 | 0.90 |
| Entropy coefficient | $1 \times 10^{-3}$ | $1 \times 10^{-2}$ |
| Value loss coefficient | 0.5 | 0.5 |
| **PACE** | | |
| Scoring function | Parameter change | Parameter change |
| Level buffer size $K$ | 1000 | 1000 |
| Replay probability $p$ | 0.7 | 0.7 |
| Staleness coefficient | 0.6 | 0.6 |
| Temperature $\beta$ | 0.8 | 0.8 |
| Minimum fill ratio $\rho$ | 0.5 | 0.5 |
| Prioritization | rank | rank |
| Exploratory gradient updates | False | True |
| Number of walls | 100(random) | – |
| **ACCEL** | | |
| Scoring function | MaxMC | PVL |
| Number of edits | 5 | 100(Noise) |
| Exploratory gradient updates | False | False |
| Number of walls | 0 | – |
| **PLR$^\perp$** | | |
| Scoring function | MaxMC | PVL |
| Exploratory gradient updates | False | False |
| Number of walls | 100(random) | – |
| **PLR** | | |
| Scoring function | L1 value loss | L1 value loss |
| Exploratory gradient updates | True | True |
| Number of walls | 100(random) | – |

