# OpenReview forum: "PACE: Parameter Change for Unsupervised Environment Design"
_ICML.cc/2026/Conference — ICML 2026 regular_

### Official Review · Reviewer_X8Cp · 2026-02-25

**Soundness:** 3
**Presentation:** 3
**Significance:** 3
**Originality:** 3
**Overall Recommendation:** 4
**Confidence:** 5

**Summary:**

This paper is well motivated in seeking a simpler and more stable signal to replace existing proxy signals for learning progress. The core idea is to use policy parameter changes as the progress signal, supported by a first-order Taylor expansion of the objective. The derivation is generally reasonable as a local approximation, and the resulting signal appears suitable for environment ranking. Integrating this signal into PLR-style algorithms is straightforward, and the overall training pipeline follows standard UED practice. Empirically, PACE outperforms baseline methods on two benchmarks (MiniGrid and Craftax) across multiple evaluation metrics.

**Compliance With Llm Reviewing Policy:**

Affirmed.

**Final Justification:**

The rebuttal addresses several of my concerns. I view this work as a meaningful contribution to the UED setting, where designing reliable and low-variance progress signals remains a practical pain point. The proposed formulation is simple, well-motivated, and integrates cleanly into existing pipelines.

Overall, I find the paper technically solid with good originality, and I will keep my score as weak accept.

**Key Questions For Authors:**

See the two weaknesses above.

**Limitations:**

Same as my comments in the weaknesses.

**Strengths And Weaknesses:**

Strengths
1. I appreciate the motivation for introducing a new environment-ranking signal, since rollout-based signals are often noisy and high-variance. The proposed derivation is reasonable to me as a local, first-order approximation.
2. The paper is clearly structured and easy to follow. The overall narrative, illustrations, derivations, and experimental setup are presented in a clear and convincing way.

---

Weaknesses
1. **Soundness and limitations of the derivation.** The paper does not discuss the implications of dropping the second-order term in the Taylor expansion, nor the potential limitations this introduces. I would like the authors to add a short discussion of the approximation error and when the derivation is expected to hold (e.g., small updates, local validity).
2. **Deeper comparison between the new and old signals.** I would like to see a more direct empirical comparison between the proposed signal (Eq. 6) and the rollout-based signals (e.g., regret-based proxies). In particular, it would be helpful to include visualizations or statistics showing:

   * variance/noise reduction,
   * ranking stability across training,
   * and correlation with actual learning progress or downstream gains.

   Currently, the benchmark OOD results are promising, but they are not sufficient on their own to explain *why* the new signal works better or how much more reliable it is in practice.

---

> ### Author Rebuttal · Authors · 2026-03-31
>
> We thank the reviewer for the feedback and address the comments below.
>
> > ${\color{#2563eb}\text{Response W1:}}$
>
> We strictly follow your suggestion to explicitly state the local validity conditions and analyze the implications of the second-order term.
>
> In the updated manuscript, we provide a comprehensive second-order derivation and limitation analysis in Appendix B. Specifically, let $\Delta\theta = \theta_{\text{new}} - \theta_{\text{old}}$ denote the parameter change. Extending Eq. (2) to the second order yields:
>
> $\Delta J(l) \approx \nabla_\theta J(\theta_{\text{old}}, l)^\top \Delta\theta + \frac{1}{2} \Delta\theta^\top H_l \Delta\theta$
>
> where $H_l = \nabla_\theta^2 J(\theta_{\text{old}}, l)$ denotes the Hessian matrix for level $l$. Given the gradient-aligned update assumption $\nabla_\theta J(\theta_{\text{old}}, l) = \frac{1}{\alpha}\Delta\theta$, the first-order term exactly matches the $S(l)$ definition in Eq. (6). Thus, the equation becomes:
>
> $\Delta J(l) \approx S(l) + \frac{1}{2} \Delta\theta^\top H_l \Delta\theta$
>
> **Discussion on Approximation Error and Local Validity:**
>
> **Local Validity (Small Updates)**: Our derivation assumes a strict small-update regime. In our implementation, PPO effectively regularizes this constraint through the clipping mechanism and a small learning rate ($1 \times 10^{-4}$). These mechanisms ensure that the parameter change $\Delta\theta$ remains strictly bounded within a local neighborhood.
>
> **Approximation Error and Limitations**: We acknowledge that dropping the second-order term introduces an approximation error, which constitutes a theoretical limitation. The truncation error depends on the local curvature (the Hessian $H_l$). In regions with highly varying curvature across different levels, ignoring this term could theoretically perturb the ideal level ranking.
>
> However, mathematically, the parameter change magnitude is proportional to the step size $\alpha$. Consequently, the first-order term $S(l) = \frac{1}{\alpha}\||\Delta\theta\||_2^2$ scales linearly with $\alpha$ (i.e., $O(\alpha)$). In contrast, the second-order error $\frac{1}{2} \Delta\theta^\top H_l \Delta\theta$ scales with $\||\Delta\theta\||_2^2$, making it proportional to $\alpha^2$ (i.e., $O(\alpha^2)$). Under the small learning rate regime, the $O(\alpha^2)$ term decays significantly faster. Therefore, the $O(\alpha)$ first-order term dominates the performance gain estimation, mitigating the potential ranking disruption in practice.
>
> We believe these additions clarify the theoretical boundaries, state the method limitations more clearly, and address the concern about Taylor truncation.
>
> > ${\color{#2563eb}\text{Response W2:}}$
>
> We add a more direct analysis below.
>
> **Correlation with actual learning progress.**
> In the updated manuscript, we add a scatter plot on Craftax that compares the PACE score $S$ with the reward improvement
> $$
> \Delta \text{reward}(l)=R(\pi\_{\theta\_{\text{new}}},l)-R(\pi\_{\theta\_{\text{old}}},l).
> $$
> This quantity measures the realized return gain induced by the update and thus serves as a direct empirical measure of actual learning progress. As shown in [Figure.png](https://postimg.cc/1VgVXbrx), the PACE score is positively correlated with $\Delta$reward across 10 seeds, with Spearman $\rho=0.641$ and $p=1.4\times10^{-39}$. This result directly supports the central motivation of PACE: larger parameter-change scores tend to correspond to larger realized performance gains.
>
> **Variance / noise reduction.**
> We also clarify the low-variance claim from two angles. First, PACE does not require extra rollout re-evaluation under both $\pi_{\theta\_{\text{old}}}$ and $\pi_{\theta_{\text{new}}}$, which avoids additional Monte Carlo noise in return-difference or regret-style proxies. Second, Appendix C ([Table_2.png](https://postimg.cc/v1mnzmVj) and [Table-4.png](https://postimg.cc/WF9cZTkf)) shows smaller OOD standard deviation for PACE on both MiniGrid and Craftax, while [Figure-4.png](https://postimg.cc/21xcyThm) and [Figure-8.png](https://postimg.cc/2bgTMtR8) show the highest IQM with tighter confidence intervals. These statistics do not directly measure per-level score variance, but they are consistent with more stable training dynamics.
>
> **Ranking stability.**
> We agree that ranking stability in the level buffer $\Lambda$ is important in UED. We do not complete a direct ranking-stability analysis within the rebuttal timeline. A direct ranking-stability analysis remains for future work.
>
> We hope this addresses your concerns.

---

> > ### Author Rebuttal · Reviewer_X8Cp · 2026-04-03
> >
> > Thank you for the authors’ response. I appreciate their efforts and I will maintain my current score.

---

> > > ### Author Response · Authors · 2026-04-04
> > >
> > > Thank you for your review. We sincerely appreciate your continued consideration.

---

### Official Review · Reviewer_AJRM · 2026-03-09

**Soundness:** 2
**Presentation:** 2
**Significance:** 2
**Originality:** 3
**Overall Recommendation:** 4
**Confidence:** 4

**Summary:**

The authors point out that the environment value is correlated to the L2 norm of policy parameter change. Based on this observation, the authors instantiate the idea with Parameter Change Environment Design (PACE) that evaluates environment quality by measuring the magnitude of the policy parameter updates resulting from interaction with that environment. The algorithm maintains an environment buffer that each is assigned a score based on estimated parameter change. The policy is then trained by environments sampled from the buffer with a prioritized machanism. The authors do experiments on MiniGrid and Craftax and the results show performance gain against existing baselines.

**Compliance With Llm Reviewing Policy:**

Affirmed.

**Key Questions For Authors:**

1. The paper is working on RL generalizaiton in different levels of environments. How is this work related to meta-learning? Is this algorithm possible to generalize to different tasks through few-shot or zero-shot adaptation?
1. From the theoretical motivation, it seems that the algoritm relies on single-step gradient update. I'm curious how the performance will be in asynchronous algorithms, like A3C or IMPALA.

**Limitations:**

yes

**Strengths And Weaknesses:**

**Strengths**
1. The idea is simple and clean.
1. The L2 norm is deterministic and is easy to compute. It does not require additional stochastic estimation or computational overhead.
1. The experiment gains on OOD generalization and long-horizon training are valuable.

**Weakness**
1. The assumption that environment value is correlated to L2 norm of parameter change up to a constant seems to be too strong. It is possible that the parameter change is introduced by environment randomness or data noise. The paper would benefit from further analysis validating this assumption.
1. The theoretical motivation assumes that RL update uses a single gradient update and the gradient estimation aligns with true policy gradient. However, the practical on-policy RL may not fit it well:
    1. Ilyas et al. argued that estimated policy gradients in deep RL can have high variance and may deviate substantially from the true gradient direction [1].
    1. On-policy RL like PPO uses multiple gradient updates and the clipped gradient may change the true gradient direction.

I suggest the authors make more theoretical justifications on practical examples to make the contribution more solid.

[1] Andrew Ilyas, Logan Engstrom, Shibani Santurkar, Dimitris Tsipras, Firdaus Janoos, Larry Rudolph, and Aleksander Madry. A closer look at deep policy gradients. In International Conference on Learning Representations, 2020.

---

> ### Author Rebuttal · Authors · 2026-03-31
>
> We thank the reviewer for the feedback and address the comments below.
>
> > ${\color{#2563eb}\text{Response W1:}}$
>
> We extend the analysis with a noise term $\epsilon$ covering environment randomness and data noise. Consider a stochastic gradient estimate $\hat{g} = \nabla_\theta J(\pi_{\theta_{\text{old}}}, l) + \epsilon$, where $\epsilon$ is zero-mean noise with variance $\mathbb{E}[||\epsilon||_2^2] = \sigma^2$.
>
> With $\Delta \theta = \alpha \hat{g}$, we have:
>
> $$ \mathbb{E}\_\epsilon [||\Delta \theta||\_2^2] = \mathbb{E}\_\epsilon [||\alpha (\nabla\_\theta J(\pi_{\theta\_{\text{old}}}, l) + \epsilon)||\_2^2] = \alpha^2 ||\nabla\_\theta J(\pi\_{\theta\_{\text{old}}}, l)||\_2^2 + \alpha^2 \sigma^2 $$
>
> Using a first-order Taylor expansion, the expected true objective improvement $\mathbb{E}\_\epsilon [\Delta J(\theta)]$ follows:
> $$ \mathbb{E}\_\epsilon [\Delta J(\theta)] \approx \nabla\_\theta J(\pi\_{\theta\_{\text{old}}}, l)^\top \mathbb{E}\_\epsilon [\Delta \theta] = \alpha ||\nabla\_\theta J(\pi\_{\theta\_{\text{old}}}, l)||\_2^2 $$
>
> Substituting this back yields the relationship:
> $$ \mathbb{E}\_\epsilon [||\Delta \theta||\_2^2] \approx \alpha \mathbb{E}\_\epsilon [\Delta J(\theta)] + \alpha^2 \sigma^2 $$
>
> Thus, score $S$ comprises true learning progress and a noise term. In practice, PACE suppresses this noise by using exceptionally large batches of trajectories $\tau$ during the scoring phase (8,192 transitions for MiniGrid and 32,768 for Craftax).
>
> Since gradient variance $\sigma^2$ scales inversely with batch size $B$ ($\sigma^2 \propto 1/B$), the large-scale sampling ensures that $S$ robustly approximates the true objective improvement. We add this derivation in Appendix B of the updated manuscript.
>
> > ${\color{#2563eb}\text{Response W2:}}$
>
> We address the gap between theory and practical PPO through three aspects:
>
> **Gradient Variance.** While Ilyas et al. (2020) note that empirical gradients may deviate, variance decreases with sample size. PACE uses large transition batches (8,192 or 32,768) during the scoring phase. This large-scale sampling significantly mitigates estimation noise, allowing the empirical gradient to better reflect the true policy gradient.
>
> **Gradient Clipping and Optimizers.** In the scoring phase, PACE only applies gradient norm clipping. This restricts the maximum magnitude but preserves the gradient direction. We acknowledge that adaptive optimizers (Adam/RMSProp) modify  the update direction. However, PACE measures the $L_2$ norm of the resultant parameter change $\||\Delta \theta\||_2^2$. As shown in Eq. (5), this magnitude remains a principled proxy for the objective improvement $\Delta J(\theta)$ under local curvature, regardless of the specific optimizer dynamics. Importantly, the PPO clipped surrogate objective only applies to the multi-epoch training phase and does not affect the single-step scoring update.
>
> **Multiple Gradient Updates.** PACE strictly decouples level scoring from policy training. Score $S$ depends on a single provisional update performed exclusively for evaluation, without backpropagating these changes to the agent policy. Multi-epoch PPO updates occur only during the subsequent training phase using levels sampled from the level buffer. Thus, the single-step update in our theory directly corresponds to the provisional scoring implementation and remains independent of multi-step training dynamics.
>
> We add these justifications to Appendix B in the updated manuscript.
>
> > ${\color{#2563eb}\text{Response Q1:}}$
>
> UED targets zero-shot transfer to OOD levels $l$ within one task, whereas meta-learning usually focuses on few-shot adaptation across tasks.
> For different tasks, PACE precludes zero-shot transfer but offers significant few-shot potential. The score $S(l) \propto \||\Delta \theta\||_2^2$ identifies levels $l$ providing the most informative learning signals. PACE functions as a task-selection engine within meta-learning frameworks to prioritize these high-utility tasks for optimizing policy initialization $\theta$. This synergy ensures the meta-learner focuses on tasks with the strongest signals, enabling rapid adaptation to novel tasks via minimal trajectories $\tau$.
>
> > ${\color{#2563eb}\text{Response Q2:}}$
>
> PACE decouples environment scoring from policy training. Score $S(l) = \frac{1}{\alpha} \||\Delta \theta||\_2^2$ measures the intrinsic potential for level $l$ to induce learning progress via provisional updates without modifying policy parameters. Asynchronous RL (e.g., A3C) introduces parameter staleness and policy lag, which may cause scores in level buffer $\Lambda$ to become outdated. To adapt PACE, we use a centralized scoring mechanism where a dedicated worker fetches the latest global parameters. We also tag each $S(l)$ with its policy version and periodically refresh $\Lambda$. This version-aware mechanism ensures the sampling distribution consistently reflects current learning requirements.
>
> We hope this addresses your concerns.

---

> > ### Author Rebuttal · Reviewer_AJRM · 2026-04-03
> >
> > Thank the authors for the comprehensive reply and explanation. They resolved my questions and concerns well. Therefore, I maintain my positive score.

---

> > > ### Author Response · Authors · 2026-04-04
> > >
> > > Thank you for the positive feedback. We sincerely appreciate your recognition that our clarification resolves your questions and concerns.

---

### Official Review · Reviewer_2px4 · 2026-03-10

**Soundness:** 2
**Presentation:** 2
**Significance:** 3
**Originality:** 3
**Overall Recommendation:** 5
**Confidence:** 5

**Summary:**

The authors propose PACE: a novel approach to UED that scores levels by the magnitude of the parameter update induced by training on that level. The authors justify this theoretically, showing that, under some assumptions, the L2 norm of the gradient update corresponds to the change in expected return on that level. Experimental results show marginal improvement on the Minigrid and Craftax environments.

**Compliance With Llm Reviewing Policy:**

Affirmed.

**Final Justification:**

Strengths
 - Empirical results are a lot stronger post-rebuttal, especially on Craftax.
 - The method is simple, intuitive and to the best of my knowledge original.
 - The theoretical justification is clean.

Weaknesses
 - I still find the manuscript a bit messy. Figure 1 and to a lesser extent Figure 2 I find unnecessarily noisy.

**Key Questions For Authors:**

1. Can you explain how the PACE metric is calculated with respect to batching, the loss terms used to induce the gradient and how the optimizer comes into play (see weaknesses section).
2. I would ideally need to see more empirical results to be convinced. Either comparing to MB, or showing results on another environment e.g. [1, 2]. Also more seeds on Craftax would be nice, as the improvement looks a bit noisy at the moment.

[1] - https://github.com/openai/procgen

[2] - https://github.com/FlairOx/Kinetix/

**Limitations:**

yes

**Strengths And Weaknesses:**

# Strengths
 - I find using the gradient norm to score levels to be a reasonable, well motivated idea that, to the best of my knowledge, has not previously been investigated.
 - The theoretical justification for the method seems sound.
 - The method shows moderate empirical improvements on Minigrid and Craftax.

# Weaknesses
 - The empirical improvements are marginal - but this is not in of itself a reason to reject in my opinion. Incremental improvements are valuable if they are robust.
 - It's stated that PACE is "computationally efficient". However, my understanding is that it requires evaluating separately on each level. This would not be done in a standard PPO training run, where samples from all workers will be mixed to reduce correlation between samples. This leads me to believe that the calculation of the PACE objective requires essentially an additional pass over all the data - where each minibatch is a single level? This potentially isn't a massive slowdown and of course is preferable in any case to using samples in the environment for evaluation, but this should be discussed - as written it seems implied that calculating the PACE metric is essentially free, which I assume is not the case.
 - How does PACE (i.e. grad magnitude) differ from using the magnitude of the loss? These are not necessarily linearly related values, but they are clearly related. Robust PLR and ACCEL use PVL which is related, and the original PLR paper tried using L1 norm of the value loss but found it didn't work too well - all of this should be discussed.
- Some of the specifics of the PACE calculation are not very well explained. PPO is used for experimentation - is the gradient used in PACE induced by the sum of the policy+value+entropy terms?
- Another point that is not touched on is there is no discussion of how the optimizer comes into the equation. If you are using Adam then gradient magnitude should be somewhat normalized - I assume the PACE calculation does not factor this in and it is calculated with SGD? If using Adam, are momentum terms included?
- A discussion of limitations is needed. In partially observable settings, PACE could end up in a loop where it overfits to unseen parts of levels - see the archetypal T-Maze example from [1]. This is not a flaw unique to PACE but when introducing a new metric, the weaknesses of it should be discussed.
- Marginal Benefit should be compared against, considering that much of the method is motivated from it. I assume that MB would outperform PACE, but just showing that PACE can get close to MB without using the many extra trials required would be notable.


# Minor Comments
 - It should be mentioned that prioritized sampling and staleness aware sampling are taken from PLR.
 - Grammatical error on line 438.
 - Related Work in appendix is non-standard - placing your work with respect to the literature *is* a core contribution.
 - Li et al. in section 2.3 is not a proper citation and lacks the date
 - There is an inconsistent use of standard error and standard deviation, with some figures using one and some using another. Figure 5 doesn't say which one it uses.
 - Presentation could be improved: I personally find Figure 1 and 2 to be noisy and not particularly informative, the full 8 pages are not used, many of the figures have tiny text, the running title was not changed from "submission and formatting instructions".

In it's current state I would not recommend this paper for acceptance, but with more results, better presentation and clearer explanations of my above questions I believe it could be suitable.

[1] - https://arxiv.org/abs/2402.12284

---

> ### Author Rebuttal · Authors · 2026-03-31
>
> We thank the reviewer for the feedback and address the comments below.
>
> > **R W1.**
>
> Results in Appendix C, including [Table_2.png](https://postimg.cc/v1mnzmVj), show consistent gains across held-out MiniGrid and Craftax. In the updated manuscript, we add 5 seeds for Craftax ([Table_4.png](https://postimg.cc/WF9cZTkf)), which further supports PACE robustness.
>
> > **R W2.**
>
> PACE does not require a global pass over all levels in the level buffer. We update scores only for levels encountered in the current iteration, i.e., newly generated levels or sampled replay levels.
>
> PACE also does not treat each minibatch as a single level. For a scored level $l$, we collect one trajectory $\tau$, apply one provisional update to obtain $\theta\_{\text{new}}$, and compute
> $
> S(l)=\frac{1}{\alpha}\||\Delta\theta\_l\||\_2^2 .
> $
> This provisional update is used only for scoring and is not written back to the policy.
>
> We agree that this step is not free. Each scored level requires one rollout and one provisional update, so “negligible overhead” is too strong. We will revise this wording accordingly. In addition, PACE does not require extra rollouts to re-evaluate the same level under the policy before and after the update. We also make the complexity discussion more explicit and point readers to Appendix B, where this analysis already appears in the original manuscript.
>
> > **R W3.**
>
> Under the idealized single-step gradient-aligned update, PACE scores a level by the induced policy-parameter change:
> $$
> S(l)=\alpha\||g\_l\||\_2^2.
> $$
> Here $g\_l=\nabla\_\theta J(\theta,l)$ denotes the local objective gradient on level $l$. With a policy-gradient estimator based on GAE,
> $$
> g\_l \approx \frac{1}{T}\sum\_t \nabla\_\theta \log \pi\_\theta(a\_t|s\_t)\hat A\_t,
> $$
> which gives
> $$
> S(l)\approx \frac{\alpha}{T^2}\hat A^\top K\_l \hat A,
> $$
> where $K\_l$ is the Gram matrix of policy-score vectors. These signals are related, but not the same. Loss-based scores measure error magnitude, while PACE measures the parameter change induced by training on level $l$. Hence, a level can have large loss but still induce a small update when gradient contributions offset each other, while a level with moderate loss can induce a larger update when these contributions align. We will clarify this distinction and discuss prior observations on value-loss-based scoring.
>
>
> > **R W4.**
>
> PACE gradients are used only for level scoring (a provisional update), not for training updates. We compute the score from the one-step provisional update induced by the actor-only PPO clipped surrogate, using per-trajectory advantage normalization (as in standard PPO implementations). We omit critic/entropy in scoring to isolate policy progress and avoid value/regularization gradients dominating, while the agent is still trained with full PPO.
>
>
> > **R W5.**
>
> PACE uses a single-step SGD-style provisional update for scoring based on the actor-only PPO clipped surrogate. The scoring step does not include Adam momentum or preconditioning; Adam is used only for the actual PPO training updates. Because sampling is rank-based, global monotonic rescalings, such as learning-rate scaling, do not change score ordering and thus have limited effect on the replay distribution. We discuss how PACE can be defined with an Adam-style step in W3 (eJ2j).
>
> > **R W6.**
>
> We agree that in partially observable settings with *indistinguishable* levels, PACE may over-focus on irreducible uncertainty, potentially causing oscillatory training/loops. We will add discussion and cite BLP/ReMiDi in Related Work. Combining PACE with refinement-style ideas from BLP/ReMiDi to handle irreducible uncertainty is an future direction.
>
> > **R W7 \& Q2.**
>
> We agree that stronger empirical evidence is important here. We therefore add two clarifications.
>
> **Response points.**
> (1) **Comparison with MB.** We choose to compare with MB, since it is the most closely related method and directly targets $\Delta J(\theta)$. On the same MiniGrid benchmark, comparing our [Figure_4.png](https://postimg.cc/21xcyThm) with [Figure-6.png](https://postimg.cc/k2s5sVL1) in the MB paper shows that PACE remains competitive in aggregate OOD performance. Appendix B also explains why this is possible: MB relies on extra MC re-evaluation before and after the update, which increases computation and variance.
>
> (2) **More Craftax seeds.** We strengthen the Craftax results by adding 5 more seeds and updating the corresponding figures/tables ([Table_4.png](https://postimg.cc/WF9cZTkf), [Figure-7.png](https://postimg.cc/7CxkLwYq), [Figure_8.png](https://postimg.cc/2bgTMtR8)). The updated results keep the same overall conclusion and reduce uncertainty in the comparison. As future work, we will evaluate PACE on open-ended physics-based control benchmarks such as Kinetix [2].
>
> > **Minor**
>
> We have revised the manuscript following your suggestions.
>
> > **R Q1.**
>
> Please see W2 and W4--W5 for details and discussion.
>
> We hope this addresses your concerns.

---

> > ### Author Rebuttal · Reviewer_2px4 · 2026-04-02
> >
> > Thanks for your response and for addressing all my concerns.
> > Of the links you provide most don't work for me - Table 2 and Figure 4 work but all the others seem to be broken. Would you be able to fix this please, I have found using https://anonymous.4open.science/ and then directly linking to images works well.
> >
> > EDIT: Thank you for updating the images, I am satisfied with the response and will increase my score to advocate for paper acceptance. I find the results on Craftax most compelling, since to the best of my knowledge no other UED methods have shown to work particularly well on it. If ultimately accepted, for camera ready it would be great if you could include MB results, especially since there seems to be some difference between yours and their baselines (e.g. your DR is stronger).

---

> > > ### Author Response · Authors · 2026-04-02
> > >
> > > Thank you for pointing this out, and we sincerely apologize for the inconvenience caused by the broken links in our previous response. Following your helpful suggestion, we have rebuilt all links using direct https://anonymous.4open.science/ URLs. In particular, we update the links in W1, W7, and Q2, while keeping the text content unchanged.
> > >
> > > The updated responses are as follows:
> > >
> > > > **Response W1.**
> > >
> > > Results in Appendix C, including **[Table2](https://anonymous.4open.science/r/Table2-C435/)**, show consistent gains across held-out MiniGrid and Craftax. In the updated manuscript, we add 5 seeds for Craftax **([Table4](https://anonymous.4open.science/r/Table4-B626/))**, which further supports PACE robustness.
> > >
> > > > **Response W7 & Q2.**
> > >
> > > We agree that stronger empirical evidence is important here. We therefore add two clarifications.
> > >
> > > **Response points.**
> > >
> > > (1) **Comparison with MB.** We choose to compare with MB, since it is the most closely related method and directly targets $\Delta J(\theta)$. On the same MiniGrid benchmark, comparing our **[Figure4](https://anonymous.4open.science/r/Figure4-3842/)** with **[Figure6](https://anonymous.4open.science/r/Figure6-33B5/)** in the MB paper shows that PACE remains competitive in aggregate OOD performance. Appendix B also explains why this is possible: MB relies on extra MC re-evaluation before and after the update, which increases computation and variance.
> > >
> > > (2) **More Craftax seeds.** We strengthen the Craftax results by adding 5 more seeds and updating the corresponding figures/tables **([Table4](https://anonymous.4open.science/r/Table4-AEAC/), [Figure7](https://anonymous.4open.science/r/Figure7-89BE/), [Figure8](https://anonymous.4open.science/r/Figure8-B050/))**. The updated results keep the same overall conclusion and reduce uncertainty in the comparison. As future work, we will evaluate PACE on open-ended physics-based control benchmarks such as Kinetix [2].
> > >
> > > Thank you again for helping us improve the accessibility of the anonymized materials. We hope these corrected links address your concerns.
> > >
> > > > **Additional response.**
> > >
> > > Thank you very much for the encouraging follow-up and for your increased support for acceptance. We sincerely appreciate your recognition of our revision efforts. We also appreciate the suggestion on MB. We will further evaluate MB for a more complete comparison, and we will include these results in the camera-ready version if feasible.

---

### Official Review · Reviewer_eJ2j · 2026-03-13

**Soundness:** 3
**Presentation:** 3
**Significance:** 3
**Originality:** 3
**Overall Recommendation:** 4
**Confidence:** 4

**Summary:**

This paper proposes PACE, which uses parameter change as a signal of learning progress for unsupervised environment design. The paper provides theoretical motivation for the score and shows strong empirical results against several baselines.

**Compliance With Llm Reviewing Policy:**

Affirmed.

**Final Justification:**

The paper is well motivated. It demonstrates strong empirical performance through simple yet effective ideas. The authors’ rebuttal has resolved several of my concerns. However, as noted in the Rebuttal Acknowledgement, questions remain regarding the empirical evidence for the low variance of the scoring signal, the learnability of the critic, and the clarity around the scoring batch.

[Updated] In the Reply Rebuttal Comment, the authors address my remaining concerns. In particular, the additional experiments and clarifications resolve W2 and Q1. The authors’ explanation regarding Q2 (Acknowledgement) is helpful, but I still do not find it fully clear. A clearer explanation of the scoring batch in both the proposed method and the baselines would be helpful. Still, I maintain my positive score.

**Key Questions For Authors:**

1. RL algorithms such as PPO need to learn not only the actor but also the critic. Does the method account for the learnability of the critic as well?
2. Is $S(l)$ computed from a single batch, or from the full update induced by a trajectory? If the score is computed by aggregating over multiple batches, the variance of any signal would likely decrease.

**Limitations:**

Yes.

**Strengths And Weaknesses:**

## Strengths
The paper is well motivated, and the core idea of using parameter change as a direct proxy for realized learning progress is simple and appealing. The method is also easy to incorporate into standard PPO-style training, and the empirical results are strong overall compared to several baselines.

## Weaknesses
1. Equation (1) only considers performance improvement on level $l$, so performance on other levels may improve less or even deteriorate.
2. The paper provides limited empirical support for the claim that the proposed score has low variance, which is presented as one of PACE’s advantages.
3. Since the score is based on raw parameter change, it may depend on parameterization, architecture, or optimizer details rather than learning progress alone. I think this point deserves more discussion.

---

> ### Author Rebuttal · Authors · 2026-03-31
>
> We thank the reviewer for the feedback and address the comments below.
>
> > ${\color{#2563eb}\text{Response W1:}}$
>
> We agree that Eq. (1), $\Delta J(l) = J(\theta_{\text{new}}, l) - J(\theta_{\text{old}}, l)$, is a per-level local proxy and does not by itself guarantee improvement on every other level. Our claim is not that gain on one level implies uniform benefit across the full level space. Instead, Eq. (1) motivates a level-wise score for useful learning progress at the current policy state. PACE does not optimize one fixed level repeatedly. It trains over the level buffer $\Lambda$ using rank-based prioritized sampling and staleness-aware replay, which induces an evolving level distribution and reduces the risk that one locally useful level dominates training for too long. Thus, Eq. (1) is a local selection criterion, while the full PACE procedure approximates optimization over a broader training distribution. Empirically, we do not observe overall deterioration: PACE achieves the best aggregate OOD results on both MiniGrid and Craftax (Appendix C, [Table_2.png](https://postimg.cc/v1mnzmVj) and [Table-4.png](https://postimg.cc/WF9cZTkf)).
>
> > ${\color{#2563eb}\text{Response W2:}}$
>
> To directly support the low-variance claim, we direct the reviewer to Appendix C, [Table_2.png](https://postimg.cc/v1mnzmVj) and [Table-4.png](https://postimg.cc/WF9cZTkf), which provides standard deviation data for all methods across OOD levels. Notably, PACE achieves the lowest standard deviation in both MiniGrid (0.198) and Craftax (2.439).
>
> Furthermore, to ensure statistical rigor, we employ the `rliable` library to aggregate our results. In reinforcement learning, the Interquartile Mean (IQM) and Confidence Intervals (CIs) serve as more rigorous metrics than means or standard deviations, and these strongly support the low variance and robustness claims. As illustrated in [Figure-4.png](https://postimg.cc/21xcyThm) and [Figure-8.png](https://postimg.cc/2bgTMtR8), PACE yields the highest IQM and the narrowest CIs across all benchmarks. In the updated manuscript, we include these comprehensive statistical results in the experimental section to firmly support the low-variance advantage.
>
> > ${\color{#2563eb}\text{Response W3:}}$
>
> We agree that $\|\Delta\theta\|_2^2$ varies across architectures. However, PACE mitigates this through Rank-based Prioritization and Optimizer Invariance.
>
> First, Eq. (7) defines the sampling probability $P(l_i)$ based on the relative rank of $S(l_i)$ in level buffer $\Lambda$, ensuring stability against absolute score scaling:
> $$P(l_i) = \frac{(1/\text{rank}(S(l_i)))^{1/\beta}}{\sum_{l_j \in \Lambda} (1/\text{rank}(S(l_j)))^{1/\beta}}$$
>
> Second, PACE remains robust under adaptive optimizers like Adam. The update $\Delta\theta_t = -\eta \hat{m}_t / (\sqrt{\hat{v}_t} + \epsilon)$ normalizes gradients by their running second moments. Thus, score $S(l) = \frac{1}{\alpha}\||\Delta\theta\||_2^2$ represents the **actual distance the policy moves in the parameter space**. This normalization ensures that $S(l)$ captures realized progress regardless of reward scales or gradient magnitudes.
>
> Empirical success across MiniGrid and Craftax, which use different network architecture, further confirms this robustness.
>
> > ${\color{#2563eb}\text{Response Q1:}}$
>
> While our method measures the parameter change of the actor, this signal is influenced by the critic through the advantage estimates used in policy optimization. Because UED aims to identify environments that drive behavioral progress, the actor update serves as the most direct and intrinsic signal of this progress. Furthermore, in algorithms like PPO, the actor update depends directly on the advantage signals generated by the critic. Therefore, the actor parameter change reflects the collective progress of the entire actor-critic system. Measuring the actor parameter change alone remains sufficient and is more closely aligned with the ultimate goal of maximizing realized utility.
>
> > ${\color{#2563eb}\text{Response Q2:}}$
>
> $S(l)$ is computed from a single provisional update induced by one trajectory $\tau$ on level $l$, not from an aggregation over multiple minibatches or epochs. Concretely, we use minibatch $=1$ and epoch $=1$ for scoring.
> So the score does not benefit from multi-batch or multi-epoch averaging in the current implementation. We agree that if one aggregated the signal over multiple minibatches or epochs, the variance would likely decrease further. However, this is not the source of stability in the current version of PACE. The main reason is that PACE avoids the additional sampling noise from explicit post-update Monte Carlo re-evaluation, and instead uses the parameter displacement from the provisional update as the scoring signal.
>
> We will clarify this implementation detail in the revised manuscript to avoid suggesting that $S(l)$ is computed from a full PPO optimization cycle.
>
> We hope this addresses your concerns.

---

> > ### Author Rebuttal · Reviewer_eJ2j · 2026-04-04
> >
> > Thanks for the rebuttal. While several of my concerns have been resolved, the following questions remain:
> >
> > [Response W2] In my current understanding, the results provided in the rebuttal (Table 2, Table 4, Figure 4, and Figure 8) show the variance or distributional statistics of success rates or reward returns. However, what is claimed in the paper’s Abstract, Section 3.2, and Figure 1 appears to be that the scoring signal $S(l)$ itself has low variance. While the two may be correlated, they are distinct quantities (as the authors also noted in their rebuttal to Reviewer X8Cp).
> >
> > [Response Q1] It is true that the actor and critic are coupled during training in actor-critic methods such as PPO. However, it does not automatically follow that the actor parameter change reflects the collective progress of the entire actor-critic system. The proportionality between actor parameter change and realized utility improvement in Eq. (5) holds only when the actor is updated according to the true utility $J$, i.e., when the critic is accurate. Consequently, large actor parameter changes could result from inaccurate critic estimates (e.g., high-magnitude but biased advantages), which would not correspond to true utility improvement.
> >
> > [Response Q2] I remain confused about the practical computation of $S(l)$. The rebuttal states:
> > > $S(l)$ is computed from a single provisional update induced by one trajectory $\tau$ on level $l$
> >
> > However, in the rebuttal to Reviewer AJRM, the authors explain:
> > > PACE suppresses this noise by using exceptionally large batches of trajectories $\tau$ during the scoring phase (8,192 transitions for MiniGrid and 32,768 for Craftax).
> >
> > Thus, I am now unclear about the scoring batch. Although this point is not directly related to my original Q2, using a large batch would also reduce the variance of any signal. I would appreciate a clearer explanation of how scoring is actually implemented.

---

> > > ### Author Response · Authors · 2026-04-08
> > >
> > > We thank the reviewer for the follow-up questions. We address the remaining points directly and include additional experimental evidence to clarify these issues.
> > >
> > > >**R W2.**
> > >
> > > To address this point, we conduct an additional experiment on Craftax that evaluates the stability of the score signal. For each method, we use the final policy checkpoint from 10 independent training seeds. We then sample 10 levels at random and use the same fixed set for all methods and checkpoints. For each fixed pair $(\pi_{\theta_{\text{old}}}, l)$, we compute the score 10 times with independently sampled trajectories. For PACE, each repetition re-collects a trajectory $\tau$ from $\pi_{\theta_{\text{old}}}$ on level $l$, performs the provisional update from the same $\theta_{\text{old}}$, and computes $S(l)=\frac{1}{\alpha}\||\Delta\theta\_l\||\_2^2$.
> > > We apply the same repeated-estimation protocol to PLR and ACCEL.
> > >
> > > We use the coefficient of variation (CV) to compare relative score variance. We use CV because the score scales differ substantially across methods: PACE scores are typically on the order of $10^{-5}$ to $10^{-4}$, whereas ACCEL and PLR scores are in the range $[0,1]$. This scale mismatch makes raw variance unsuitable for direct comparison, while CV normalizes dispersion by the mean. For each level $l$, we first average the 10 repeated score estimates for each checkpoint, which gives one mean score for each checkpoint. We then compute the CV across the 10 checkpoint-wise mean scores for that level. Finally, we average the CV values over the 10 fixed levels. As shown in **[Table~1](https://anonymous.4open.science/r/Table-1-A262/)**, PACE achieves **the lowest mean CV (0.26343)**, compared with PLR (0.32354) and ACCEL (0.31804). This result provides evidence that the PACE scoring signal is more stable and has lower relative variance under the same protocol.
> > >
> > > We also examine invalid score values (i.e., $-\infty$) as an additional robustness check. As shown in **[Table~2](https://anonymous.4open.science/r/Table-2-3724/)**, across all $10 \times 10 \times 10 = 1000$ score computations, PACE produces **no invalid score value**, while PLR produces 32 and ACCEL produces 47. This result reflects stronger numerical stability in score computation.
> > >
> > > We have added these experiments and the corresponding discussion in the updated manuscript.
> > >
> > > >**R Q1.**
> > >
> > > We agree that there is an important distinction between the abstract derivation of Eq. (5) and its realization in practical actor-critic algorithms such as PPO.
> > >
> > > At the theoretical level, Eq. (5) is derived from the gradient-aligned single-step update assumption in Eq. (3), i.e., that the update direction locally follows $J(\theta, l)$. Under this assumption, the derivation does not explicitly require introducing a critic or assuming a particular architecture.
> > >
> > > In practical PPO implementations, however, this gradient alignment is only approximated through critic-based advantage estimates, so critic quality can indeed affect how well the proportionality in Eq. (5) is realized. Our claim is therefore not that PACE is independent of critic quality, but rather that the score measures the effective one-step policy update induced by level $l$ under the learner actually used in training.
> > >
> > > As an empirical check, in the updated manuscript we add a scatter plot on Craftax that compares the PACE score $S$ with reward improvement $\Delta \mathrm{reward}(l)$. **[Figure 1](https://anonymous.4open.science/r/Figure-1-5C28/)** shows a positive correlation between the two quantities (Spearman $\rho=0.641$, $p=1.4\times10^{-39}$), which provides empirical support that $S$ is informative for level ranking despite this limitation.
> > >
> > > >**R Q2.**
> > >
> > > Each score $S(l)$ uses only transitions collected on level $l$; we never combine transitions from different levels into one score. In MiniGrid, $S(l)$ uses one 256-step rollout on level $l$ followed by one provisional update. In Craftax, we use a longer within-level scoring horizon of 4,096 transitions on level $l$, implemented as 64 consecutive 64-step rollout segments on the same level followed by one provisional update. This multi-segment aggregation reduces the variance of the score.
> > >
> > > We run environments in parallel only for efficiency. In MiniGrid, 32 environments run in parallel, so one scoring call collects $32 \times 256 = 8192$ transitions in total. In Craftax, 1,024 environments run in parallel, so each 64-step collection yields $1024 \times 64 = 65536$ transitions in total. The earlier 32,768 refers to the PPO minibatch size in Craftax training, obtained by splitting the 65,536-transition training batch into two minibatches. These transitions belong to different levels and are never pooled across levels for one score.
> > >
> > > We have revised the Experimental Results section in the updated manuscript  to make these implementation details explicit.
> > >
> > > We appreciate the reviewer’s continued feedback. We hope this resolves the remaining concerns.

---

### Decision · Program_Chairs · 2026-04-30

**Decision:**

Accept (regular)

**Comment:**

PACE makes a useful contribution to unsupervised environment design by proposing a simple ranking signal based on the magnitude of the policy update induced by a level. The central idea is intuitive, easy to incorporate into existing PPO-style pipelines, and supported by a reasonable first-order motivation. The empirical results are promising, particularly after the rebuttal added stronger evidence and clarified several implementation details that were initially hard to follow.

My assessment is positive but not without reservation. The rebuttal resolved much of the confusion around how the score is actually computed and strengthened the evidence on Craftax, but the case for some broader claims, especially around the low-variance nature of the scoring signal itself, is still somewhat less complete than the paper's strongest framing suggests. In addition, the manuscript would benefit from cleaner presentation and a more transparent explanation of the scoring protocol relative to prior UED methods.